# Strikingly different neurotransmitter release strategies in dopaminergic subclasses

Ana Dorrego-Rivas*†, Darren J Byrne†, Yunyi Liu, Menghon Cheah, Ceren Arslan, Marcela Lipovsek‡, Marc C Ford, Matthew S Grubb*

Centre for Developmental Neurobiology, Institute of Psychiatry, Psychology and Neuroscience (IoPPN), King's College London, London, United Kingdom

**\*For correspondence:**
ana.dorrego-rivas@kcl.ac.uk
(AD-R);
matthew.grubb@kcl.ac.uk (MSG)

†These authors contributed
equally to this work

**Present address:** ‡Ear Institute,
University College London,
London, United Kingdom

Reviewing Editor: Jun Ding,
Stanford University, Stanford,
United States

## eLife Assessment

This study provides evidence for distinct neurotransmitter release modalities between two subclasses of dopaminergic neurons in the olfactory bulb. Specifically, it demonstrates dendritic neurotransmitter release in anaxonic neurons and axonal release in axon-bearing neurons. The presence of GABAergic self-inhibition in anaxonic neurons further underscores the functional divergence between these subtypes. Overall, the manuscript presents **solid** evidence and offers biologically **important** insights into the organization and function of dopaminergic circuits within the olfactory bulb.

**Abstract** Neuronal function is intimately tied to axodendritic polarity. Neurotransmitter release, for example, is usually the role of the axon. There are widespread exceptions to this rule, however, including many mammalian neuronal types that can release neurotransmitter from their dendrites. In the mouse olfactory bulb, closely related subclasses of dopaminergic interneuron differ markedly in their polarity, with one subtype lacking an axon entirely. These axon-bearing and anaxonic dopaminergic subclasses have distinct developmental profiles and sensory responses, but how their fundamental polarity differences translate to functional outputs remains entirely unknown. Here, we provide anatomical evidence for distinct neurotransmitter release strategies among these closely related dopaminergic subtypes: anaxonic cells release from their dendrites, while axon-bearing neurons release exclusively from their intermittently myelinated axon. These structural differences are linked to a clear functional distinction: anaxonic, but not axon-bearing, dopaminergic neurons are capable of self-inhibition. Our findings suggest that variations in polarity can produce striking distinctions in neuronal outputs, and that even closely related neuronal subclasses may play entirely separate roles in sensory information processing.

## Introduction

Neurons are one of the most striking examples of cell polarisation. Most neurons have canonical axodendritic polarity, defined by the presence of a somatodendritic domain and a single axon (*Banker, 2018*). The morphological and molecular specialisation of these distinct compartments gives them unique functional properties, with dendrites usually receiving most synaptic inputs and axons sending outputs to other neurons by releasing neurotransmitters. However, exceptions to this neuronal polarity dogma are common. In the mammalian brain, there are neurons – including midbrain dopaminergic (DA) cells (*Pucak and Grace, 1994*; *Falkenburger et al., 2001*) and cortical bitufted cells (*Zilberter et al., 1999*) – that release neurotransmitter from their dendrites. There are also neurons – including

retinal amacrine cells (*Wu et al., 2011*; *Goaillard et al., 2020*) – that lack canonical axons altogether. A remarkable hub for such unconventional neuronal polarity is the olfactory bulb (OB), the first part of the brain that processes information about the sense of smell. Dendritic release of neurotransmitters is a widespread phenomenon in the OB, even in projection neurons and other cells with classically defined axodendritic polarity (*Schoppa and Urban, 2003*; *Hayar et al., 2004*; *Imamura et al., 2020*). In addition, non-canonical polarity is extremely common in the OB, where almost all GABAergic inter-neurons lack an axon entirely (*Tufo et al., 2022*).

Neuronal polarity in the OB can be completely different even in closely related neuronal subtypes. In the glomerular layer, a major class of interneurons – around 90,000 cells total in the mouse OB (*McLean and Shipley, 1988*; *Panzanelli et al., 2007*; *Parrish-Aungst et al., 2007*) – releases both dopamine and GABA in order to modulate neurotransmitter release from olfactory sensory neuron terminals and to regulate downstream processing in local circuits (*Wachowiak and Cohen, 1999*; *Berkowicz and Trombley, 2000*; *Borisovska et al., 2013*; *Liu et al., 2013*; *Banerjee et al., 2015*; *Vaaga et al., 2017*; *Garg et al., 2025*). We recently found that these DA neurons can be classified into two subtypes depending on a binary morphological distinction: the presence or absence of an axon. This fundamental divergence in polarity in anaxonic and axon-bearing DA cells is correlated with different excitability profiles and odour response properties (*Chand et al., 2015*; *Galliano et al., 2018*; *Lau et al., 2024*). It also corresponds closely to previous descriptions of OB DA heterogeneity based on cell size (*Kosaka and Kosaka, 2008*; *Pignatelli and Belluzzi, 2017*), with axon-bearing DA neurons having significantly larger somas (*Kosaka and Kosaka, 2008*; *Galliano et al., 2018*). Whilst axon-bearing DA neurons represent the minority in terms of numbers, making up ~2.5% of the OB's DA population (*Galliano et al., 2018*), they have the potential to significantly influence olfactory processing via inter-glomerular inhibition (*Kosaka and Kosaka, 2008*; *Liu et al., 2013*; *Whitesell et al., 2013*; *Banerjee et al., 2015*). However, the specific function of these two neuron subtypes within OB glomerular networks is not fully understood. A crucial outstanding question is how the structural differences in these two neuron subtypes translate into strategies for influencing bulbar circuits via neurotransmitter release.

Anaxonic DA neurons only have a soma and dendrites (*Chand et al., 2015*; *Galliano et al., 2018*), making these the sole available domains for neurotransmitter release. Indeed, dendritic presynaptic structures are present in DA neurons (*Kiyokage et al., 2017*) and are associated with these cells' functional ability to self-inhibit via local GABA release (*Smith and Jahr, 2002*; *Murphy et al., 2005*; *Maher and Westbrook, 2008*). On the other hand, axon-bearing DA neurons are known to be the sole source of long-range interglomerular connections in the OB (*Kosaka and Kosaka, 2008*), with functional studies showing that these cells release neurotransmitters over these long-distance axonal projections (*Liu et al., 2013*; *Whitesell et al., 2013*; *Banerjee et al., 2015*). However, in a brain region where dendritic release is the norm rather than the exception, what remains unknown is whether axon-bearing DAs combine this distal, *inter*glomerular signalling with local, *intra*glomerular inhibition via release from their dendrites. Answering this question has important implications for how information is processed in early olfactory circuits.

Here, we provide anatomical evidence showing that axon-bearing OB DA neurons have axons with neurotransmitter release sites, while their dendrites lack these sites almost entirely. On the other hand, we found that all anaxonic DA neurons have release sites on their dendrites. These structural differences are associated with a striking functional distinction: while all anaxonic neurons can self-inhibit, there is an almost total absence of self-inhibition responses in the axon-bearing subpopulation. This polarity-based dichotomy suggests that even these closely related interneuron subtypes may play entirely separate roles in the processing of sensory information.

## Results

### Labelling putative presynaptic structures in individual OB DA neurons

We first developed a structural approach for labelling putative presynaptic sites in individual OB DA neurons. We observed that synaptophysin, a membrane protein located in vesicles at the presynapse, is expressed in OB DA cells at both the mRNA (*Figure 1A*) and protein (*Figure 1B*) levels. Given the high synaptic density and complexity of OB glomeruli, it is not possible to reliably assign specific synaptophysin label to individual DA neurons using widespread staining of endogenous protein. We

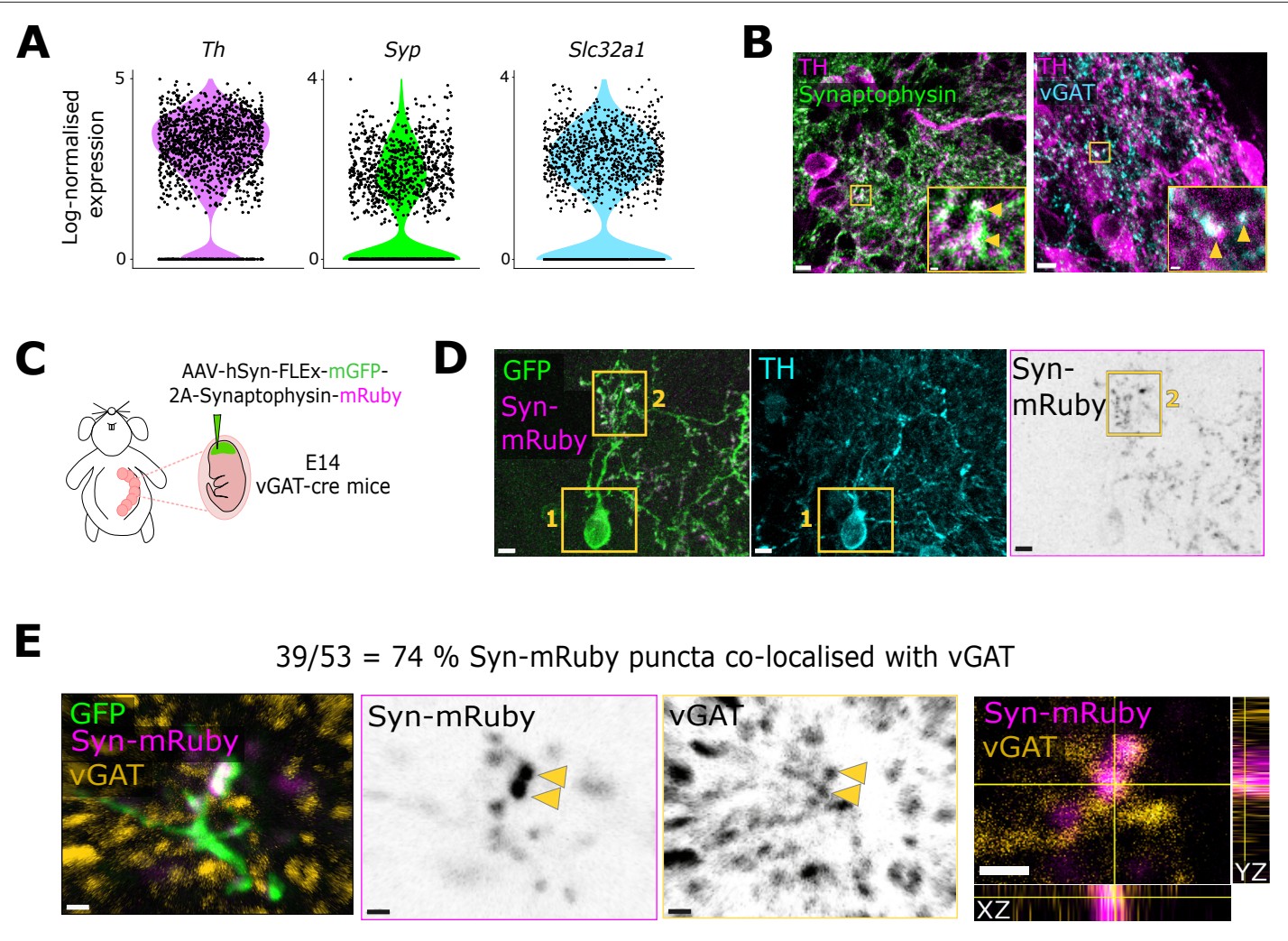

**Figure 1.** Labelling putative presynaptic structures in individual olfactory bulb (OB) dopaminergic (DA) neurons. (**A**) Log$_2$-normalised expression of *Th* (tyrosine hydroxylase), *Syp* (Synaptophysin), and *Slc32a1* (vGAT) mRNA in DA neurons from single-cell RNA sequencing data (*Brann et al., 2020*). (**B**) Example confocal images of endogenous immunostaining for synaptophysin (green) and TH (magenta) on the left, and vGAT (blue) and TH (magenta) on the right. Both images were taken in the glomerular layer of the OB. Yellow arrowheads point to small clusters where TH and synaptophysin or TH and vGAT co-localise. Scalebars: 5 μm (main) and 0.5 μm (inset) for the images on the left, 4 μm (main) and 1 μm (inset) for the images on the right. (**C**) Strategy to label putative presynaptic release sites in individual DA neurons. (**D**) Example confocal image of a successfully labelled DA cell. Inset 1 reveals the TH+ DA identity of the neuron (cyan) and inset 2 highlights the Syn-mRuby puncta (magenta, black). Scalebars: 5 μm. (**E**) Example confocal image of a GFP+ (green), Syn-mRuby+ (magenta, black) neuronal process co-stained with vGAT (orange, black). Yellow arrowheads indicate examples where Syn-mRuby and vGAT puncta co-localise. The last panel shows the orthogonal views of the bottom punctum. Scalebars: 1 μm.

therefore chose a sparse labelling approach based on the injection of an AAV-DJ-hSyn-FLEx-mGFP-T2A-Synaptophysin-mRuby construct (*Beier et al., 2015*; *Kathe et al., 2022*). This vector allows the expression in individual cre-expressing cells of a membrane-bound GFP for cell morphology, and fluorophore-tagged full-length synaptophysin for the visualisation of putative presynaptic puncta. There is strong evidence showing that the overexpression of fluorophore-tagged synaptophysin is functionally benign (*Granseth et al., 2006*; *Li and Tsien, 2012*).

We performed in utero injections in embryonic day 14 (E14) vGAT (vesicular GABA transporter)-cre mice. Since OB DA neurons are also GABAergic (*Kosaka and Kosaka, 2008*; *Kosaka and Kosaka, 2016*; *Maher and Westbrook, 2008*; *Borisovska et al., 2013*; *Vaaga et al., 2017*; *Liu et al., 2013*) and express vGAT at both the mRNA (*Figure 1A*) and protein (*Figure 1B*; *Panzanelli et al., 2007*) levels, this conditional line gives access to both anaxonic and axon-bearing DA subclasses, alongside many non-dopaminergic OB GABA-releasing neurons (*Galliano et al., 2018*). DA neurons were then

specifically identified via immunostaining against tyrosine hydroxylase (TH). TH is the rate-limiting enzyme in catecholamine synthesis and, because of the absence of noradrenergic neurons in the OB, is a specific and comprehensive marker for all DA cells in this brain region (*Hökfelt et al., 1975*; *Rosser et al., 1986*; *Kosaka and Kosaka, 2016*; *Galliano et al., 2018*).

We chose an embryonic timepoint for viral targeting to ensure the effective and sparse labelling of axon-bearing and anaxonic DA neurons (*Figure 1C*). Early infection before cells have migrated to their final positions produced sparse labelling which, when analysed in juvenile postnatal animals, permitted the unambiguous identification of an individual neuron's processes. Embryonic targeting also had the benefit of equally sampling DA subclasses. While anaxonic DA neurons are born throughout life, axon-bearing DA cells are exclusively generated in embryonic development (*Batista-Brito et al., 2008*; *Galliano et al., 2018*). Embryonic injections therefore ensured effective and more balanced labelling of both populations.

We were able to detect well-defined synaptophysin-mRuby puncta along the processes of TH+ GFP+ neurons (*Figure 1D*). To assess whether this method labels reliable presynaptic release sites in DA neurons, we co-stained the injected tissue against vGAT and observed high co-localisation between endogenous vGAT and mRuby label (*Figure 1E*; 39/53=74% synaptophysin-mRuby puncta co-localised with vGAT; n=9 cells from N=4 mice). These results demonstrate that a large proportion of synaptophysin-mRuby puncta are likely to reflect putative presynaptic release sites in OB DA neurons.

## Anaxonic DA neurons have dendritic neurotransmitter release sites

Given that anaxonic DA neurons lack an axon and are known to release both GABA and dopamine, to be able to release neurotransmitter they must have presynaptic release sites on their dendrites. We identified this neuronal subtype in our population of GFP+/TH+ neurons by the lack of the proximal axonal marker TRIM46 (*Figure 2A*, *van Beuningen et al., 2015*; *Galliano et al., 2018*).

As predicted, we found that all anaxonic neurons had synaptophysin-mRuby puncta in their dendrites (*Figure 2B and C*; n=9 anaxonic DA cells from N=4 mice). The thin slices necessary for puncta detection precluded any comprehensive analysis of dendritic morphology or synaptic distributions. The resultant sub-sampling of individual cells' arbours may also have contributed to the variability in puncta density we observed across neurons (*Figure 2C*), although similarly high variability between OB DA cell presynaptic specialisations has been shown in previous ultrastructural analyses (*Kiyokage et al., 2017*). We saw no consistent evidence for spatial patterns, clusters, or privileged branches; instead, puncta were distributed throughout each cell's dendritic arbour (*Figure 2C*). These results provide anatomical evidence for the presence of dendritic neurotransmitter release sites in anaxonic DA neurons.

## Axon-bearing DA neurons have release sites on their intermittently myelinated axons, but not on their dendrites

Long-range GABA release has been functionally demonstrated in DA neurons (*Liu et al., 2013*; *Whitesell et al., 2013*; *Banerjee et al., 2015*), strongly suggesting that axon-bearing DA neurons have axonal release sites. It was not possible to follow TRIM46+ DA axons for long distances from the soma in thin brain slices, so instead we used the distal axon marker myelin basic protein (MBP). In the first description of myelination in this cell type, we found that DA axons are myelinated in an intermittent pattern, a common trait in other brain regions (*Figure 3A*, *Tomassy et al., 2014*). Using MBP as an indicator of axon identity, we were then able to locate synaptophysin-mRuby puncta in unmyelinated sections of GFP+ TH+ axons (*Figure 3B* and *Figure 3—figure supplement 1*; n=5 axon-bearing DA cells from N=3 mice). As predicted, axon-bearing DA neurons therefore have putative presynaptic release sites on their axons.

Most neurons with canonical axodendritic polarity in the OB release neurotransmitters from both axons and dendrites (*Castro and Urban, 2009*). To assess if this scenario also holds true for DA axon-bearing cells, we explored the dendritic localisation of synaptophysin-mRuby-labelled presynaptic release sites in GFP+ neurons with clear TRIM46+ proximal axonal domains (*Figure 3C*). As expected, these axon-bearing OB DA cells had larger somas than their anaxonic neighbours (*Figure 3D* and *Figure 3—figure supplement 2A*; soma area: 110.3 ± 7.72 μm$^2$ for axon-bearing DA cells, 61.99 ± 4.41 μm$^2$ for anaxonic DA neurons; n=11 axon-bearing cells and n=9 anaxonic neurons from N=5 mice; unpaired t-test with Welch's correction, p<0.0001). Given previous findings that axon-bearing DA cells

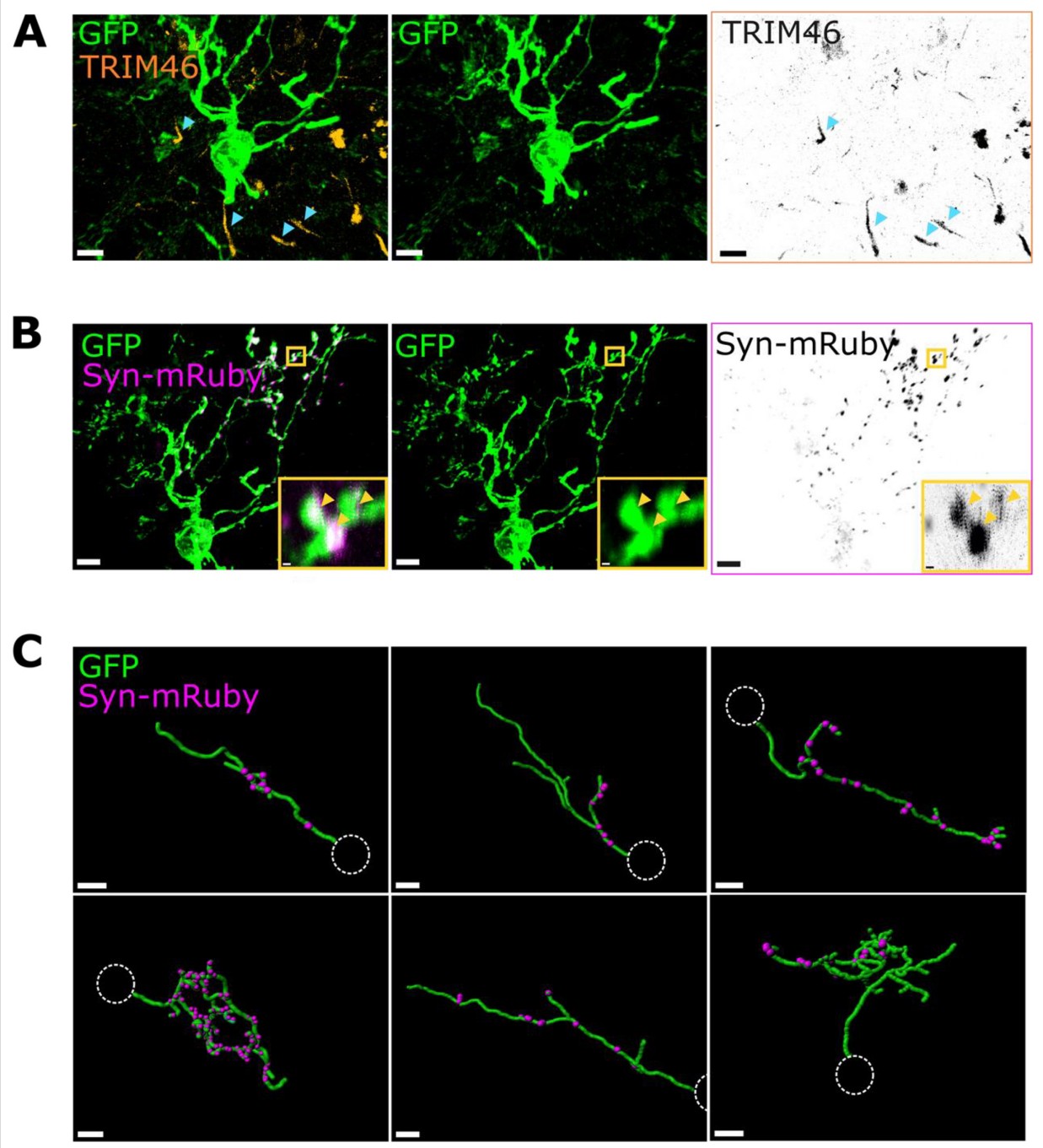

**Figure 2.** Anaxonic dopaminergic (DA) neurons have dendritic neurotransmitter release sites. (**A**) Example confocal image of a TRIM46-negative anaxonic DA neuron. Blue arrowheads point to examples of other TRIM46+ AISs (orange, black) in the same region which do not co-localise with this neuron's GFP signal. Scalebars: 5 µm. (**B**) Snapshot of the same neuron in (**A**) showing Synaptophysin-mRuby puncta (magenta, black) on the dendrites. Yellow inset highlights a region of the neuron with multiple mRuby+ puncta within the GFP+ (green) processes (yellow arrows). Scalebars: 5 µm and 0.5 µm. (**C**) Example snapshots from three-dimensional (3D) dendritic reconstructions (green, GFP) and presynaptic puncta detection (magenta, Syn-mRuby) of anaxonic DA neurons. Note: these are not full dendritic reconstructions, but example dendrites. Dotted white circle represents the soma location. Scalebars: 5 µm.

have more widely ramified dendritic trees (*Galliano et al., 2018*), we ensured that our reconstructions of TRIM46-negative dendritic processes were equally extensive in both cell subtypes (*Figure 3—figure supplement 2B*; maximum length of reconstructed dendrites: 67.64 ± 8.96 µm for axon-bearing DA cells, 74.88 ± 11.68 µm for anaxonic DA neurons; n=11 axon-bearing cells, and n=9 anaxonic neurons

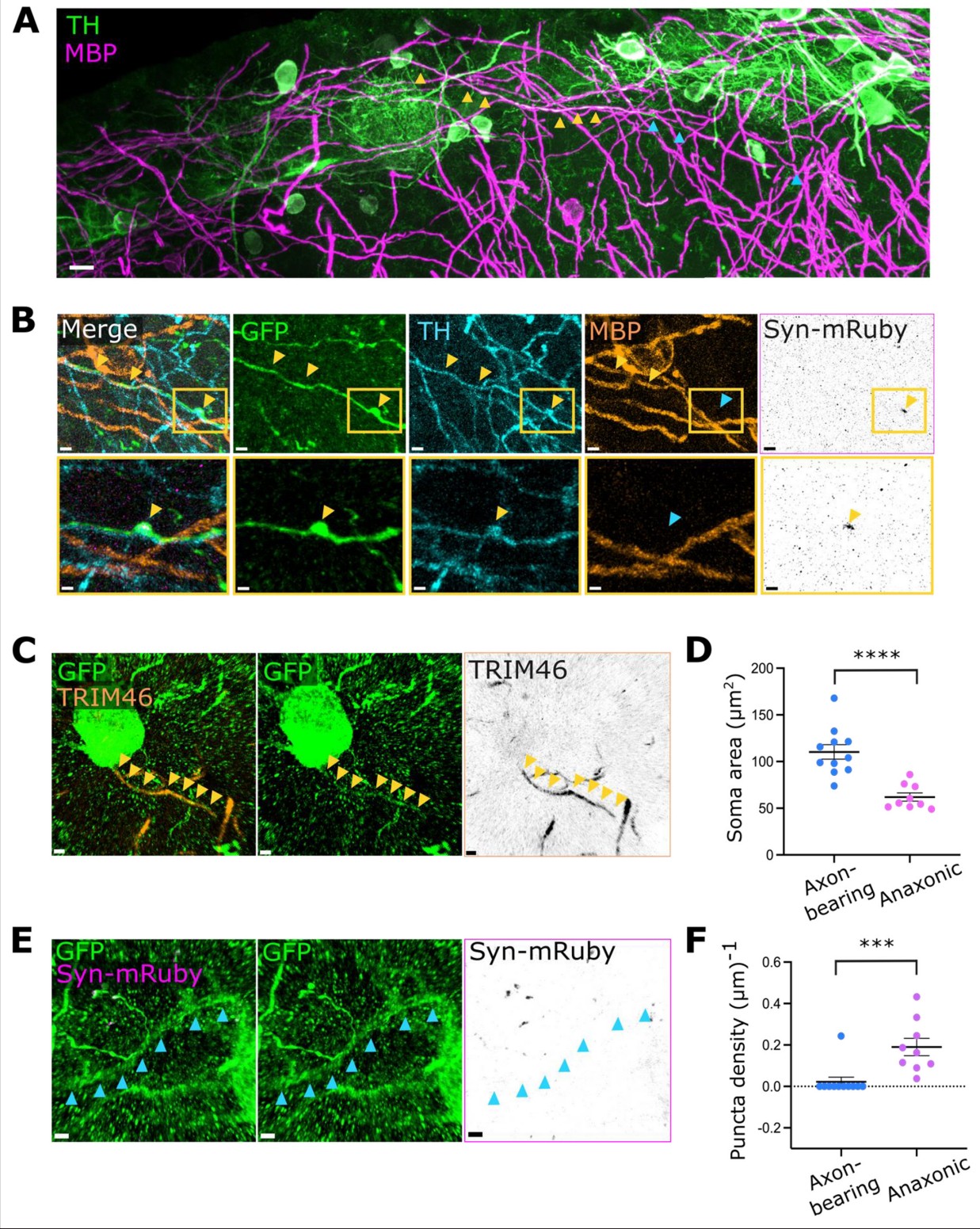

**Figure 3.** Axon-bearing dopaminergic (DA) neurons have release sites on their intermittently myelinated axons, but not on their dendrites. (**A**) Stitching of individual confocal stacks processed for maximum intensity projections of olfactory bulb (OB) DA neurons co-stained with tyrosine hydroxylase (TH) (green) and myelin basic protein (MBP, magenta). Yellow arrowheads point to myelinated parts of the axon, blue arrowheads show unmyelinated areas. Scalebar: 10 µm. (**B**) Example confocal images of a distal DA axon stained with GFP (green), TH (cyan), MBP (orange), and synaptophysin-mRuby (magenta, black). Yellow inset highlights the location of the presynaptic bouton. Yellow arrowheads point to co-localised regions, blue arrowheads

*Figure 3 continued on next page*

*Figure 3 continued*

show non-co-localisation. Scalebars: 2 µm and 1 µm. (**C**) Confocal image of an axon-bearing TRIM46+ DA neuron. Yellow arrowheads show co-localised staining for GFP (green) and TRIM46 (orange, black). Scalebars: 2 µm. (**D**) Soma area of axon-bearing and anaxonic DA neurons. Each dot shows one cell; lines show mean ± SEM; n=11 axon-bearing cells and n=9 anaxonic neurons from N=5 mice; unpaired t-test with Welch's correction; ****, p<0.0001. (**E**) Snapshot of the same axon-bearing DA neuron shown in (**C**), co-stained with GFP (green) and synaptophysin-mRuby (magenta, black). Blue arrows show dendritic segments lacking mRuby label, despite the presence of clear mRuby+ puncta in neighbouring processes from a different GFP+ cell. Scalebars: 2 µm. (**F**) Dendritic puncta density in axon-bearing and anaxonic DA neurons. All conventions as in **D**; n=11 axon-bearing cells and n=9 anaxonic neurons from N=5 mice; Mann-Whitney test; ***, p=0.0001.

The online version of this article includes the following figure supplement(s) for figure 3:

**Figure supplement 1.** Example confocal images of dopaminergic axons from different mice.

**Figure supplement 2.** Axon-bearing DA neurons have larger somas than anaxonic cells, with comparable analysed dendrite lengths and a single outlier showing dramatically high over-expressed levels of syn-mRuby puncta in the soma and dendrites.

from N=5 mice; unpaired t-test with Welch's correction, p=0.6299). We found that, unlike their anaxonic counterparts, almost all TRIM46+ neurons (10/11 cells from N=4 mice,=90.9 %) did not have any synaptophysin-mRuby puncta on their dendrites (*Figure 3E and F*; dendritic puncta density: 0.02 ± 0.02 (µm)$^{-1}$ for axon-bearing cells; 0.19 ± 0.04 (µm)$^{-1}$ for anaxonic neurons; n=11 axon-bearing cells and n=9 anaxonic neurons from N=5 mice; Mann-Whitney test; p=0.0001). We did find one single axon-bearing DA neuron with apparent dendritic puncta; however, this cell had dramatically higher levels and unusual patterns of expression as compared to all other GFP+ neurons (*Figure 3—figure supplement 2C and D*) and we include it in our sample only through an abundance of caution. These findings provide a structural basis for dendritic neurotransmitter release in anaxonic DA neurons, but not in the axon-bearing population of DA neurons in the OB.

## Anaxonic DA neurons are capable of GABAergic self-inhibition

The fact that dendritic release sites are likely located near dendritic receptors (*Kiyokage et al., 2017*) might have the functional consequence that anaxonic DA cells can signal to themselves and influence their own activity. We found that OB DA neurons do not express DA receptors at the mRNA level (*Figure 4A*), making it unlikely that they could influence their own activity via DA signalling. However, GABAergic auto-evoked inhibition (AEI) or self-inhibition has been described in OB periglomerular cells in general, and in DA neurons in particular (*Smith and Jahr, 2002*; *Murphy et al., 2005*; *Maher and Westbrook, 2008*; *Borisovska et al., 2013*).

To address if the anaxonic subtype of DA neuron can self-inhibit, we used the DAT-tdT mouse line to specifically target DA neurons (*Bäckman et al., 2006*; *Madisen et al., 2009*; *Galliano et al., 2018*; *Byrne et al., 2022*; see Materials and methods) and performed whole-cell patch clamp recordings in tdT+ cells to evoke AEI responses. We identified the anaxonic DA population by their characteristic 'monophasic' phase-plane plot profile (*Figure 4B*; *Galliano et al., 2018*). After evoking transmitter release with transient depolarisation, we observed a prolonged inward current which was blocked by adding the GABA-A receptor antagonist gabazine at 10 µM (*Figure 4C*; AEI charge: 44.11 ± 6.02 pC; n=31 cells from N=18 mice). These data reveal that anaxonic DA neurons display AEI responses, consistent with the co-existence of presynaptic release sites and postsynaptic targets in the dendrites of these cells.

Although the AEI response is initiated by the depolarisation of the recorded anaxonic DA neuron, it has always been assumed, but not demonstrated, that the response is the direct result of GABA released from the same cell. However, given the presence of other GABAergic cells in local glomerular circuits (*Kiyokage et al., 2017*; for review see *Tufo et al., 2022*), there are other potential indirect ways in which depolarising a DA neuron might produce a GABA response in that same cell.

DA neurons in the OB release GABA and dopamine (*Borisovska et al., 2013*), and D1-like and D2-like dopamine receptors are expressed in the glomerular layer (*Coronas et al., 1997*). When we depolarise an anaxonic DA neuron to evoke neurotransmitter release, it is therefore possible that the dopamine that is co-released with GABA activates the dopamine receptors of neighbouring neurons, evoking secondary GABA release that then drives the AEI response in the recorded cell (*Figure 4D*). To rule out this possibility, we showed that despite a non-significant trend towards a reduction in charge, the AEI response was still clearly present after the application of D1-like and D2-like receptor antagonists (10 µM SKF 83566-hydrobromide and 10 µM sulpiride, respectively; *Figure 4E and F*;

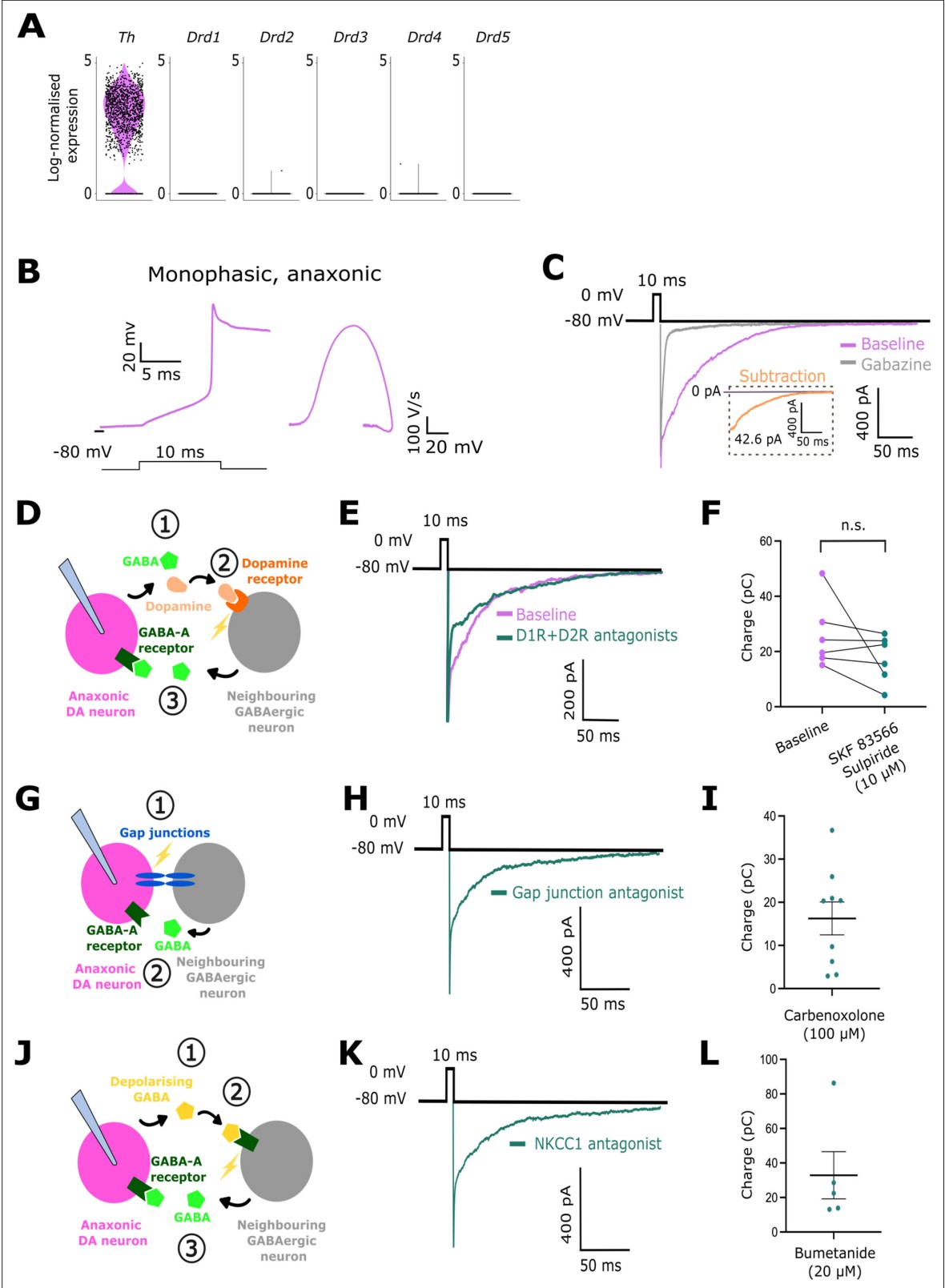

**Figure 4.** Anaxonic dopaminergic (DA) neurons are capable of self-inhibition. (**A**) Log$_2$-normalised expression of *Th*, *Drd1*, *Drd2*, *Drd3*, *Drd4*, and *Drd5* mRNA in olfactory bulb (OB) DA cells. (**B**) Example trace of an action potential fired by a putative anaxonic DA neuron (left) and its monophasic phase-plane plot profile (right). Note the prolonged repolarisation due to Cs-based internal solution. (**C**) Example traces of an auto-evoked inhibition (AEI) response recorded before (magenta) and after (grey) the application of gabazine. The subtraction is shown in the orange inset trace. (**D**) Schematic

*Figure 4 continued on next page*

*Figure 4 continued*

showing the potential involvement in the AEI response of neighbouring GABAergic neurons activated by dopamine released from the patched DA cell. (**E**) Example traces of an AEI response before (purple) and after (green) applying D1-like and D2-like receptor blockers (SR 95531 hydrobromide and sulpiride, each at 10 μM). (**F**) AEI charge before (purple) and after (green) applying dopamine receptor antagonists; n=6 cells from N=4 mice; paired t-test, p=0.21, n.s.=non-significant. (**G**) Schematic showing the potential involvement in the AEI response of neighbouring GABAergic neurons activated via gap junctions. (**H**) Example trace of an AEI response in the presence of the gap junction blocker carbenoxolone at 100 μM. (**I**) AEI charge in the presence of carbenoxolone at 100 μM. Each dot shows one cell; lines show mean ± SEM; n=9 cells from N=4 mice. (**J**) Schematic showing the potential involvement in the AEI response of neighbouring GABAergic neurons activated by depolarising GABA released from the patched cell. (**K**) Example trace of an AEI response in the presence of the NKCC1 blocker bumetanide at 20 μM. (**L**) AEI charge in the presence of bumetanide at 20 μM. All conventions as in **F**; n=5 cells from N=3 mice.

The online version of this article includes the following figure supplement(s) for figure 4:

**Figure supplement 1.** Comparison of the decay constant k of the auto-evoked inhibition response before (magenta) and after (green) the application of D1 and D2 receptor antagonists.

**Figure supplement 2.** The auto-evoked inhibition response charge is not affected by performing three repeats of the protocol or by switching the holding voltage to -50 mV.

AEI charge before blockers: 25.93 ± 4.99 pC, charge after blockers: 17.37 ± 3.47 pC; n=6 cells from N=4 mice; paired t-test, p=0.21). We also saw no significant alteration in AEI decay kinetics after the application of dopamine receptor antagonists (*Figure 4—figure supplement 1*; decay constant k before blockers: 0.03 ± 0.003 ms$^{-1}$; k after blockers: 0.03 ± 0.006 ms$^{-1}$; n=6 cells from N=4 mice; Wilcoxon matched-pairs test, p=0.84). These results show that the AEI response is not mediated by dopamine acting through other neurons.

Glomerular neurons are electrically coupled via gap junctions (*Schoppa and Westbrook, 2002*; *Kosaka et al., 2005*; *Banerjee et al., 2015*). To exclude the possibility that the electrical activation of neighbouring, GABA-releasing neurons is mediating the AEI response (*Figure 4G*), we showed that AEI was still present after gap junctions were pharmacologically blocked with 100 μM carbenoxolone (*Figure 4H and I*; AEI charge after blockers: 16.27 ± 3.83 pC; n=9 cells from N=4 mice).

Finally, another source of activation of neighbouring GABA-releasing neurons could be GABA itself: GABA can have depolarising effects on glomerular layer neurons (*Parsa et al., 2015*). To show that the GABA released by the recorded neuron is not activating neighbouring GABA-releasing cells (*Figure 4J*), we recorded strong AEI responses when depolarising GABA responses were prevented in the presence of the Na-K-Cl cotransporter 1 (NKCC1) blocker bumetanide used at 20 μM (*Figure 4K and L*; AEI charge after blockers: 32.86 ± 13.66; n=5 cells from N=3 mice). Altogether, these data robustly demonstrate that the AEI response in anaxonic DA neurons indicates true self-inhibition.

## Axon-bearing DA neurons do not self-inhibit

We next wanted to assess whether axon-bearing DA neurons can self-inhibit. In light of our anatomical data showing the presence of release sites exclusively on the distal axon, we predicted that these cells do not exert synaptic inputs onto themselves. Using the same DAT-tdT mouse line to label DA neurons, we identified tdT+ axon-bearing DA neurons by their characteristic 'biphasic' action potential waveform (*Figure 5A*, *Galliano et al., 2018*; *Galliano et al., 2021*; *Tufo et al., 2025*). Consistent with axon-bearing neurons having larger soma sizes than their anaxonic counterparts (*Kosaka and Kosaka, 2008*; *Galliano et al., 2018*; *Figure 3D*, *Figure 3—figure supplement 2A*), we found that axon-bearing DA neurons exhibited significantly lower input resistance and a non-significant trend towards higher membrane capacitance compared to anaxonic DA neurons (*Figure 5—figure supplement 1A and B*; input resistance: 324.0 ± 31.93 MΩ for axon-bearing DA neurons, n=11 cells from N=8 mice; 683.8 ± 47.11 MΩ for anaxonic DA cells, n=45 cells from N=32 mice, p<0.0001, Mann-Whitney test; membrane capacitance: 46.08 ± 4.79 pF for axon-bearing DA neurons, n=11 cells from N=8 mice; 41.57 ± 1.35 pF for anaxonic DA cells, n=45 cells from N=32 mice, p=0.50, Mann-Whitney test).

As predicted, GABA-mediated AEI currents were lacking in almost all axon-bearing DA neurons (*Figure 5B and C*; 8/9 cells from N=5 mice,=89%; mean ± SEM AEI charge for axon-bearing neurons: 3.8 ± 2.8 pC; n=9 cells from N=5 mice; anaxonic cells: 44.11 ± 6.02 pC; n=31 cells from N=18 mice; Mann-Whitney test, p<0.0001) despite these neurons having functional GABA-A receptors as shown by the presence of prominent gabazine-sensitive spontaneous IPSCs (*Figure 5D*; n=9 axon-bearing

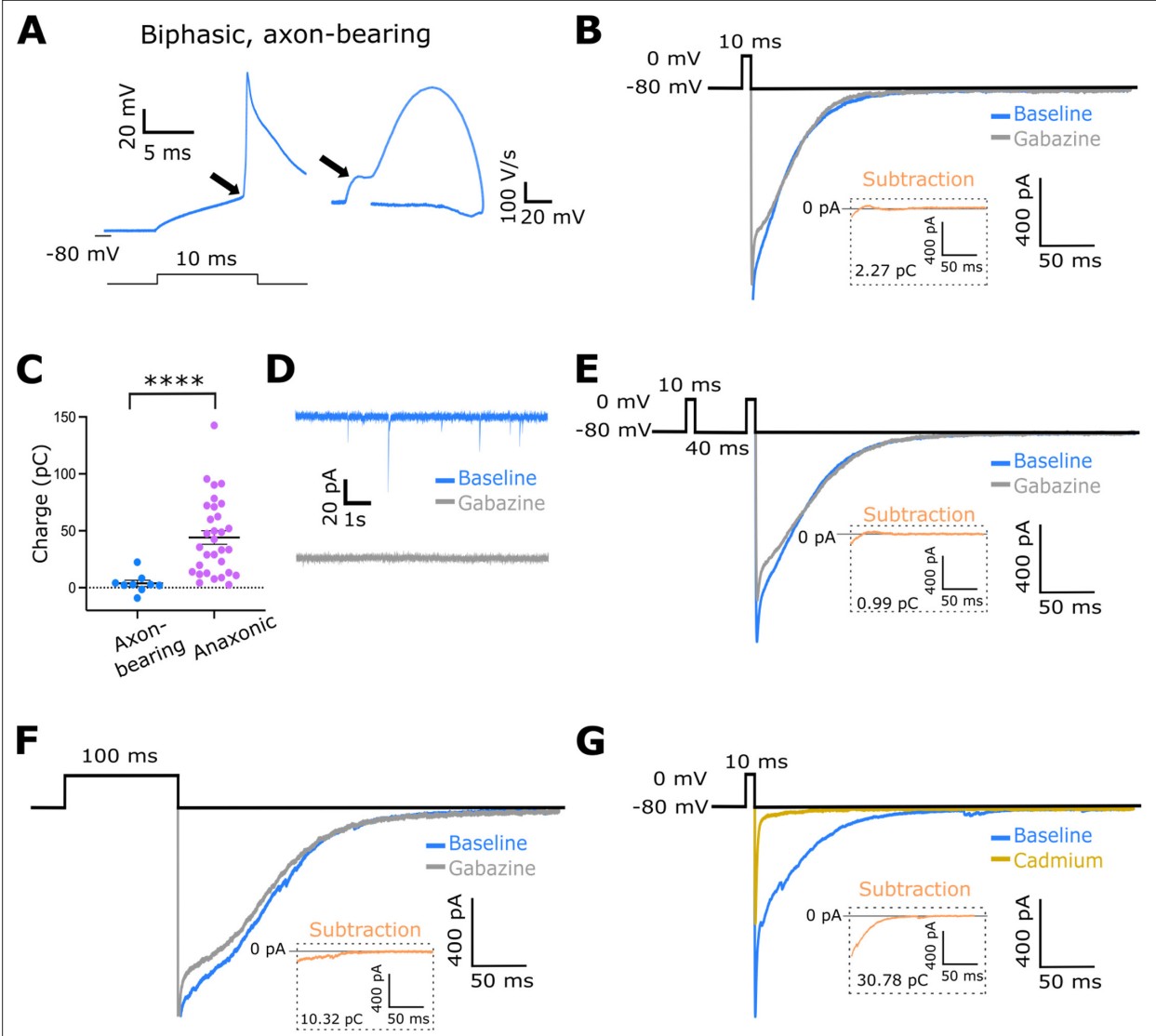

**Figure 5.** Axon-bearing dopaminergic (DA) neurons do not self-inhibit. (**A**) Example trace of an action potential fired by a putative axon-bearing DA neuron (left) and its biphasic phase-plane plot profile (right). Arrows point to spike onset kink (left) and initial segment bump (right). Note the prolonged repolarisation due to Cs-based internal solution. (**B**) Example traces showing tail current responses in an axon-bearing DA neuron before (blue) and after (grey) the application of gabazine. The subtraction is shown in the orange inset trace. (**C**) Auto-evoked inhibition (AEI) charge in axon-bearing and anaxonic DA neurons. Each dot shows one cell; lines show mean ± SEM; n=9 axon-bearing cells, n=31 anaxonic cells from N=18 mice; Mann-Whitney test; ****, p<0.0001. (**D**) Example traces showing spontaneous IPSCs in an axon-bearing DA neuron before (blue) and after (grey) the application of gabazine. (**E, F**) Example traces showing tail current responses in axon-bearing DA neurons before (blue) and after (grey) the application of gabazine using a paired-pulse protocol (**E**) and prolonged depolarisation (**F**). Subtractions are shown in the orange inset traces. (**G**) Example traces showing that the tail current recorded in axon-bearing DA neurons (blue) is blocked by cadmium (ocre). The subtraction is shown in the orange inset trace.

The online version of this article includes the following figure supplement(s) for figure 5:

**Figure supplement 1.** Passive membrane properties reflect soma-size differences between axon-bearing and anaxonic DA neurons.

neurons from N=5 mice; n=31 anaxonic cells from N=18 mice). Even when we attempted to maximise AEI amplitude via a paired-pulse protocol (***Bouhours et al., 2011***) or prolonged depolarisation, we still saw no evidence of self-inhibition in this cell subtype (***Figure 5E and F***, ***Figure 5—figure supplement 1C and D***; AEI charge for single response: 5.4 ± 2.61 pC; n=8 cells from N=7 mice; AEI charge for paired response: 1.51 ± 2.91 pC; n=8 cells from N=7 mice; Wilcoxon matched-pairs signed-rank test, p=0.06; AEI charge for 10 ms response: 0.26 ± 0.21 pC, n=7 cells from N=6 mice; AEI charge for 100 ms response: 7.75 ± 7.51 pC; n=7 cells from N=6 mice; Wilcoxon matched-pairs signed-rank test, p=0.30). Axon-bearing DA neurons did produce a tail current in response to the AEI

protocol; however, this was insensitive to gabazine and was instead abolished by adding the voltage-gated calcium channel blocker cadmium at 200 µM (*Figure 5G* and *Figure 5—figure supplement 1E*; average cadmium-sensitive current: 15.33 ± 6.95 pC, n=3 cells from N=3 mice).

In conclusion, these findings provide both structural and functional evidence showing that anaxonic DA neurons in the OB can signal locally, even to the point of self-inhibition. In contrast, axon-bearing DA neurons can – unlike all other major cell types in this brain region – only release neurotransmitters distally from their axon.

## Discussion

Our results show that different polarity-defined DA subtypes in the OB use distinct anatomical compartments for neurotransmitter release. While anaxonic DA neurons release from the dendrites, axon-bearing DA cells release nearly exclusively from the axon. These anatomical findings correlate strongly with different function: anaxonic DA neurons can self-inhibit, but this ability is absent in almost all of the axon-bearing DA population.

### The link between neuronal polarity and neurotransmitter release

Our anatomical data indicate that anaxonic DA neurons have release sites on their dendrites, while the presynaptic sites of almost all the axon-bearing DA cells are located exclusively on their intermittently myelinated axons. These findings are entirely consistent with previous descriptions of release site localisation in similar OB cell types. Early ultrastructural work showed the absence of release sites in the dendrites of axon-bearing neurons in the glomerular layer, defined morphologically as 'short-axon cells' (*Pinching and Powell, 1971*). It was also shown that axon-bearing DA neurons located in the external plexiform layer do not have dendritic release sites (*Liberia et al., 2012*). In contrast, further ultrastructural work revealed the presence of release sites in the dendrites of small DA periglomerular neurons, presumably anaxonic (*Kiyokage et al., 2017*). Our results agree with all of these findings while specifically assigning different neurotransmitter release strategies to axon-bearing and anaxonic DA neurons.

We have shown that anaxonic DA neurons use their only available anatomical compartment, the dendrites, to release neurotransmitters. This paradigm of anaxonic structural and functional polarity is unconventional but is found in other neuron types, including amacrine cells in the retina and OB granule cells (*Wu et al., 2011*; *Marshak, 2016*; *Nunes and Kuner, 2018*). In fact, dendritic release is not an uncommon phenomenon in the brain and is observed even in many axon-bearing neurons. DA neurons in the midbrain, for example, are one of the most studied examples of dendritic release (*Cheramy et al., 1981*; *Hikima et al., 2021*). In the OB, most neurons with axodendritic polarity can release from both the axon and the dendrites, including mitral cells and external tufted cells (*Isaacson and Strowbridge, 1998*; *Schoppa and Urban, 2003*; *Hayar et al., 2004*). In this context, axon-bearing OB DA neurons are unique: they do not have dendritic release despite their DA nature, and despite their identity as OB neurons with classic axodendritic polarity.

OB DA neurons co-release both GABA and dopamine (*Wachowiak and Cohen, 1999*; *Berkowicz and Trombley, 2000*; *Borisovska et al., 2013*; *Liu et al., 2013*; *Vaaga et al., 2017*; *Banerjee et al., 2015*), so do their distinct release strategies apply to both transmitters? Our anatomical approach reliably detects GABA presynaptic sites, with a high percentage of co-localisation between synaptophysin-mRuby and vGAT (*Figure 1E*). However, not all synaptophysin-mRuby puncta are vGAT-positive. This could reflect technical limitations in release site detection (*Risher et al., 2015*; *Heise et al., 2016*), but also indicates that vGAT-negative spots may represent specific dopamine release sites (*Borisovska et al., 2013*; *Liu et al., 2013*; *Vaaga et al., 2017*). Indeed, in other brain regions, synaptophysin-mRuby puncta have been used to reveal sites of dopamine release (*Li et al., 2005*). In OB DA neurons, GABA and dopamine release are associated with different functions (*Lyons-Warren et al., 2023*) and the two transmitters are known not to be released from the same vesicles (*Borisovska et al., 2013*). However, this does not necessarily imply that they are released from different active zones. Addressing this question in the current study was unfortunately not possible due to notorious difficulties in staining the dopamine vesicular transporter 2 (VMAT2) due to the lack of effective commercially available antibodies (*Zhang et al., 2015*; *Jain et al., 2023*). Whether anaxonic and axon-bearing DA

neurons release dopamine and GABA from the same sites remains an important question for future studies.

In light of our findings, many fascinating questions about anaxonic DA neurons remain to be explored. How is dendritic release developed and maintained throughout the life of the neuron? Specific mechanisms have been identified that drive the establishment and maintenance of neuronal polarity in neurons with a classic axodendritic configuration (for reviews, see *Rasband, 2010*; *Lalli, 2014*; *Bentley and Banker, 2016*). Do some of these molecular events, like cytoskeleton reorganisation and/or trafficking processes, differ in the dendrites of anaxonic neurons versus the 'standard' dendrites of axon-bearing neurons? Future work will answer these and further questions on the developmental and functional intricacies of unconventional neurons in the brain.

## Does dendritic release equal self-inhibition?

Self-inhibition in the OB was initially discovered in a subset of periglomerular cells (*Smith and Jahr, 2002*) and later described specifically in DA neurons (*Maher and Westbrook, 2008*). Here, we find that self-inhibition is not a universal OB DA neuron feature, but instead is an almost exclusive property of anaxonic DA cells. By ruling out potential external sources of GABA in the glomerular circuit, we have also provided the clearest evidence to date for true self-inhibition in OB periglomerular cells.

The link between dendritic release of neurotransmitters and self-inhibition exists in other brain regions as well. Neurons in the substantia nigra release dopamine that binds to auto-receptors and subsequently inhibits the neuron (*Pucak and Grace, 1994*; *Falkenburger et al., 2001*). Other examples include noradrenergic neurons in the locus coeruleus and serotonergic cells in the nucleus raphe dorsalis (*Aghajanian et al., 1977*; *Liu et al., 2005*). However, while these neurons have a classic axodendritic polarity and are able to release from their axons as well, anaxonic DA cells constitute a distinct model as dendrites are the solely available compartment for such function.

Finally, does dopamine released from an anaxonic OB DA neuron also modulate that same neuron? Our own data demonstrate that the AEI current is purely GABAergic: it is suppressed after the application of gabazine, but unaffected after blocking dopamine receptors (*Figure 4C, E, and F*). In addition, our transcriptomic data clearly show that OB DA neurons do not express dopamine receptors at all (*Figure 4A*). It is therefore unlikely that there is a DA component to the self-inhibition response. Previous reports of dopamine-driven responses in OB DA neurons (*Liu, 2020*) may instead have occurred via well-described gap junction connectivity between DA cells and dopamine-responsive ETCs (*Liu et al., 2013*; *Banerjee et al., 2015*).

## Implications for information processing

Anaxonic DA neurons release neurotransmitters from their dendrites, which are confined to local areas of the glomerular layer. In contrast, axon-bearing DA cells release from their axons, which spread for long distances across the glomerular layer (*Kosaka and Kosaka, 2008*; *Galliano et al., 2018*). Our findings lead to the prediction of a clear division of circuit function for very closely related cell subtypes: anaxonic DA neurons take part in local, intraglomerular inhibition, while axon-bearing DA cells participate in distal, interglomerular inhibition.

Functionally, intraglomerular inhibition acts as a 'high-pass filter' for weak odour signals detected in a glomerulus, enhancing the contrast between different olfactory stimuli (*Shao et al., 2012*; *Gire and Schoppa, 2009*; *Zak and Schoppa, 2021*). Anaxonic DA neurons are therefore expected to contribute to intraglomerular inhibition via different pathways which have been generally described for periglomerular cells. This includes the exclusively intraglomerular presynaptic inhibition of olfactory sensory neurons (*Hsia et al., 1999*; *Ennis et al., 2001*; *McGann et al., 2005*; *Korshunov et al., 2017*; *McGann, 2013*; *Vaaga et al., 2017*), as well as the dendrodendritic inhibition of multiple neighbouring glomerular cell types, including interneurons and projection neurons (*Murphy et al., 2005*; *Gire and Schoppa, 2009*). We observed heterogeneity in the puncta density along the dendrites of anaxonic DA neurons which, potential technical considerations notwithstanding, suggests that dendritic neurotransmitter release is not uniform across these cells. This variability may reflect differences in spatial positioning within the glomeruli, potentially aligning synaptic output with local circuit demands and individual postsynaptic partner types.

Self-inhibition would be expected to decrease the capacity of anaxonic DA neurons to release GABA over time, therefore reducing inhibitory influence within glomerular circuits. Self-inhibition

could therefore control the timing of intraglomerular inhibition, allowing subsequent stimuli to activate glomerular networks. Self-inhibition could also aid the high-pass filter function of glomerular networks (*Gire and Schoppa, 2009*). If it occurs exclusively in the context of strong DA neuron activation, it could decrease intraglomerular inhibition to promote glomerular output when input strength goes past a certain threshold. There is also the possibility that self-inhibition is more of a bug than a feature. Perhaps if anaxonic DA neurons have to release GABA from their dendrites in order to perform their intraglomerular circuit functions, the chances are high that their nearby postsynaptic receptors cannot escape some of that GABAergic influence. Further work is needed to investigate how these mechanisms influence overall intraglomerular inhibition and glomerular output.

Glomeruli in the OB create a spatial map of odour identity, where different odours specifically activate different combinations of glomeruli (*Soucy et al., 2009*; *Murthy, 2011*). Interglomerular inhibition acting between different glomeruli selectively enhances the signal contrast in between individual stimuli, boosting odour identification and/or discrimination (*Uchida et al., 2000*; *Urban, 2002*). This circuit function must be exclusively undertaken by axon-bearing DA neurons, due to their unique capacity for long-range axonal release of neurotransmitters (this paper; *Kosaka and Kosaka, 2008*; *Liu et al., 2013*; *Whitesell et al., 2013*; *Banerjee et al., 2015*).

Several key traits are shared across the entire population of OB DA neurons, including their DA and GABAergic identities and their glomerular location in the olfactory circuitry. Despite these similarities, we have found fundamental differences between OB DA subtypes in terms of their anatomy, neurotransmitter release compartments and the presence/absence of self-inhibition. These are strong foundations to suggest entirely different sensory processing functions in even very closely related interneuron sub-populations.

## Materials and methods

### Key resources table

| Reagent type (species) or resource | Designation | Source or reference | Identifiers | Additional information |
|---|---|---|---|---|
| Strain, strain background (*Mus musculus*) | C57BL/6J mice | Charles River | Strain code 027; RRID:MGI:3028467 | |
| Strain, strain background (*M. musculus*) | DAT-Cre, B6.SJL-*Slc6a3*tm1.1(cre)Bkmn/J | The Jackson Laboratory | Jax stock 006660; RRID:IMSR_JAX:006660 | |
| Strain, strain background (*M. musculus*) | VGAT-Cre, *Slc32a1*tm2(cre)Lowl/J | The Jackson Laboratory | Jax stock 016962; RRID:IMSR_JAX:016962 | |
| Strain, strain background (*M. musculus*) | Flex-tdTomato, B6.Cg–Gt(ROSA)26Sortm9(CAG-tdTomato)Hze/J | The Jackson Laboratory | Jax stock 007909; RRID:IMSR_JAX:007909 | |
| Antibody | Mouse monoclonal Anti-Tyrosine Hydroxylase | Merck | Catalogue number MAB318; RRID:AB_2201528 | Used at 1:500 1:5000 1:10,000 1:50,000 |
| Antibody | Chicken polyclonal Anti-GFP | Abcam | Catalogue number ab13970; RRID:AB_300798 | Used at 1:1000 |
| Antibody | Guinea pig polyclonal Anti-TRIM46 | Synaptic Systems | Catalogue number 377005; RRID:AB_2721101 | Used at 1:500 |
| Antibody | Rabbit polyclonal Anti-vGAT | Synaptic Systems | Catalogue number 131002; RRID:AB_887871 | Used at 1:1000 |
| Antibody | Rat monoclonal Anti-mCherry (16D7) | Thermo Fisher Scientific | Catalogue number M11217; RRID:AB_2536611 | Used at 1:500 |
| Antibody | Rabbit polyclonal Anti-MBP | Thermo Fisher Scientific | Catalogue number PA5-78397; RRID:AB_2736178 | Used at 1:1000 |
| Antibody | Rat monoclonal Anti-MBP (clone 12) | Merck | Catalogue number MAB386; RRID:AB_94975 | Used at 1:500 |
| Antibody | Guinea pig polyclonal Anti-Synaptophysin1 | Synaptic Systems | Catalogue number 101 004; RRID:AB_1210382 | Used at 1:500 |
| Antibody | Alexa Fluor 488, Goat Anti-Chicken IgY (H+L) secondary antibody | Invitrogen | Catalogue number A-11039; RRID:AB_2534096 | Used at 1:1000 |

*Continued on next page*

*Continued*

| Reagent type (species) or resource | Designation | Source or reference | Identifiers | Additional information |
|---|---|---|---|---|
| Antibody | Alexa Fluor 488, F(ab')2-Goat Anti-Mouse IgG (H+L) secondary antibody | Invitrogen | Catalogue number A-11017; RRID:AB_ 2534084 | Used at 1:1000 |
| Antibody | Alexa Fluor 488, Goat Anti-Guinea Pig IgG (H+L) secondary antibody | Invitrogen | Catalogue number A-11073; RRID:AB_ 2534117 | Used at 1:1000 |
| Antibody | Alexa Fluor 647, Goat Anti-Mouse IgG (H+L) secondary antibody | Invitrogen | Catalogue number A-21235; RRID:AB_ 2535804 | Used at 1:1000 |
| Antibody | Alexa Fluor 488, Goat Anti-Rabbit IgG (H+L) secondary antibody | Invitrogen | Catalogue number A-11008; RRID:AB_ 143165 | Used at 1:1000 |
| Antibody | Alexa Fluor 546, Goat Anti-Rabbit IgG (H+L) secondary antibody | Invitrogen | Catalogue number A-11010; RRID:AB_2534077 | Used at 1:1000 |
| Antibody | Alexa Fluor 594, Goat Anti-Rat IgG (H+L) secondary antibody | Invitrogen | Catalogue number A-11007; RRID:AB_10561522 | Used at 1:1000 |
| Transfected construct (Adeno-associated virus) | AAV-DJ-hSyn-FLEx-mGFP-T2A-Synaptophysin-mRuby | Stanford Vector Core Facility | Reference GVVC-AAV-100 | Used at ~$3 \times 10^{12}$ GC/ml |
| Chemical compound, drug | DL-2-Amino-5-phosphonopentanoic acid (APV) | Sigma-Aldrich | Catalogue number A5282 | Used at 50 µM |
| Chemical compound, drug | 2,3-Dioxo-6-nitro-7-sulfamoyl-bnzo[f]quinoxaline (NBQX) | Sigma-Aldrich | Catalogue number N183 | Used at 10 µM |
| Chemical compound, drug | Tetrodoxin citrate (TTX) | Alomone and Tocris | Catalogue numbers T-550 and 1069 | Used at 1 µM |
| Chemical compound, drug | SR 95531 hydrobromide (Gabazine) | Tocris | Catalogue number 1262 | Used at 10 µM |
| Chemical compound, drug | SKF 83566 hydrobromide | Tocris | Catalogue number 1586 | Used at 10 µM |
| Chemical compound, drug | Sulpiride | Sigma-Aldrich | Catalogue number S8010 | Used at 10 µM |
| Chemical compound, drug | Cadmium chloride | Sigma-Aldrich | Catalogue number 202908 | Used at 200 µM |
| Chemical compound, drug | Carbenoxolone disodium | Tocris | Catalogue number 3096 | Used at 100 µM |
| Chemical compound, drug | Bumetanide | Tocris | Catalogue number 3108 | Used at 20 µM |
| Software, algorithm | R package Seurat | The R Project for Statistical Computing | RRID:SCR_016341 | |
| Software, algorithm | ImageJ software (Fiji) | NIH; *Schindelin et al., 2012* | RRID:SCR_003070 | |
| Software, algorithm | Prism 10 | GraphPad | RRID:SCR_002798 | |
| Software, algorithm | Patchmaster v2x73 | HEKA Elektronik | RRID:SCR_000034 | |
| Software, algorithm | MATLAB R2024b | MathWorks | RRID:SCR_01622 | |
| Software, algorithm | Imaris v9.1.2 | Oxford Instruments | RRID:SCR_007370 | |
| Software, algorithm | ImarisViewer (v10.2.0) | Oxford Instruments | RRID:SCR_027660 | |
| Software, algorithm | Zen 2010 B-2P1 | Zeiss | RRID:SCR_013672 | |
| Software, algorithm | Inkscape v1.3.2 | Inkscape | RRID:SCR_014479 | |

## Experimental design

Sample sizes were not established through a priori power calculations because no predefined effect size informed their selection. Instead, we performed several independent experimental runs incorporating multiple biological replicates. Information regarding sample size and statistical tests is provided in the figure legends and in the 'Data analysis' section. Data acquisition and evaluation were conducted by several investigators, who were blinded to experimental conditions whenever feasible. Randomisation of acquisition/recording order in axon-bearing and anaxonic cells was not feasible because the axon-bearing neurons were rarer in the tissue and more fragile during patch-clamp. To maximise yield, we prospectively prioritised recording/acquiring the axon-bearing neurons when available. When both populations were concurrently available and stable, we alternated acquisition order to avoid systematic ordering effects.

## Animals

We used mice of either sex and housed them under a 12 hr light-dark cycle in an environmentally controlled room with access to water and food ad libitum. Wild-type C57BL/6J mice (Charles River) were used either as experimental animals or to back-cross each generation of transgenic animals. The founders of our transgenic mouse lines – vGAT-cre (Slc32a1[tm2(cre)Lowl]/J, Jax stock 016962), DAT-cre (B6. SJL-Slc6a3[tm1.1(cre)Bkmn]/J, Jax stock 006660), and flex-tdT(B6.Cg-Gt(ROSA)26Sor[tm9(CAG-tdTomato)Hze]/J, Jax stock 007909) – were purchased from The Jackson Laboratory. Immunohistochemistry experiments with vGAT-cre mice were performed at postnatal day (P)28. Patch-clamp electrophysiology experiments with DAT-tdT mice were performed between P21 and P35. All experiments were performed in accordance with the United Kingdom Animals (Scientific Procedures) Act 1986 and were approved by the King's College London Animal Welfare and Ethical Review Body and the United Kingdom Home Office (Home Office Project licence: PP9786830).

## Viral injections

To sparsely label neuronal morphology together with presynaptic release sites, vGAT-cre mice were injected in utero via intraventricular injection with AAV-DJ-hSyn-FLEx-mGFP-T2A-Synaptophysin-mRuby ($3 \times 10^{12}$ GC/ml, AAV DJ GVVC-AAV-100, Stanford Vector Core Facility) (**Beier et al., 2015**; **Kathe et al., 2022**) at E14. Time-mated pregnant mice were anaesthetised with isoflurane, used at 4% for general anaesthesia induction and at 2% for anaesthesia maintenance. Injections (≈1 µl) were performed with a glass-pulled pipette (1B150-F4, World Precision Instruments) through the uterine wall into one of the lateral ventricles of the embryos. The uterine horns were then returned into the abdominal cavity, the wall and the skin were sutured, and the embryos were allowed to continue their development in utero and born naturally until experimental use. Post-surgery, mice recovered in a 32°C chamber and were housed individually.

## Immunohistochemistry

P28 and P33 mice were anaesthetised with an overdose of pentobarbital and then perfused with 20–25 ml of PBS with heparin (A16198, Thermo Fisher Scientific) (20 units/ml), followed by 20–25 ml of 4% paraformaldehyde (PFA; P001, TAAB Laboratories) in 3% sucrose, 60 mM PIPES, 25 mM HEPES, 5 mM EGTA, and 1 mM $MgCl_2$ (all purchased from Sigma-Aldrich). The OBs were dissected and post-fixed in 4% PFA overnight (4°C), then embedded in 5% agarose and sliced at 50 µm using a vibratome (VT1000S, Leica). Free-floating slices were washed with PBS 1× and permeabilised and blocked in PBS with 5% normal goat serum (NGS, G9023, Sigma-Aldrich) and 0.25% Triton (Triton X-100, Sigma-Aldrich) for 2 hr at room temperature (RT). Slices were then incubated in primary antibody solution (antibodies diluted in PBS with 5% and 0.25% Triton) for 2–5 days at 4°C. Primary antibodies, working dilutions, and sources are indicated in *Table 1*. Slices were then washed three times for 10 min with PBS, before being incubated in secondary antibody solution (species-appropriate Life Technologies Alexa Fluor-conjugated; all at a 1:1000 dilution in PBS with 5% NGS and 0.25% Triton) for 2 hr at RT. After washing three times for 10 min with PBS, slices were mounted on glass slides (J2800AMNZ, Epredia) with FluorSave (345789, Merck).

**Table 1.** Primary antibodies.

| Target | Host | Dilution | Supplier | Reference | Identifier |
|---|---|---|---|---|---|
| TH | Mouse | 1:500 1:5000 1:10,000 1:50,000 | Merck | MAB318 | RRID:AB_2201528 |
| GFP | Chicken | 1:1000 | Abcam | ab13970 | RRID:AB_300798 |
| TRIM46 | Guinea pig | 1:500 | Synaptic Systems | 377005 | RRID:AB_2721101 |
| vGAT | Rabbit | 1:1000 | Synaptic Systems | 131002 | RRID:AB_887871 |
| mCherry | Rat | 1:500 | Thermo Fisher | M11217 | RRID:AB_2536611 |
| MBP | Rabbit | 1:1000 | Thermo Fisher | PA5-78397 | RRID:AB_2736178 |
| MBP | Rat | 1:500 | Millipore | MAB386 | RRID:AB_94975 |
| Synaptophysin | Guinea pig | 1:500 | Synaptic Systems | 101004 | RRID:AB_1210382 |

All immunostainings involving multiple targets were performed by incubating the primary antibodies simultaneously, except for the combination of mouse anti-TH and rat anti-MBP, where a sequential incubation was necessary to avoid cross-reactivity (first TH, then MBP).

## Image acquisition, analysis, and illustration

### Acquisition

All images were acquired using a laser scanning confocal microscope (Zeiss LSM 710, equipped with Zen 2010 B-SP1 software) using appropriate excitation and emission filters, a pinhole of 1 AU and 40× or 63× oil immersion objectives. Laser power and gain were set to prevent fluorescence saturation and allow acquisition with an optimal signal-to-noise ratio, and z-stacks were acquired at software-defined optimal thickness and step sizes. For puncta detection and dendritic reconstruction, stacks were acquired with 1× zoom (0.033 µm/pixel), 2048 × 2048 pixels, and in z-stacks with 0.37 µm steps. The DA identity of the cells was assessed qualitatively by taking a single plane snapshot of the TH and GFP channels that included the soma. Analysed cells were located exclusively in the OB's glomerular layer, including the border with the external plexiform layer. Axon-bearing neurons displayed at least one clear elongated TRIM46-positive region (of approximately 10–25 µm in length) that belonged to a GFP+ process. For co-localisation analyses, stacks were acquired with 2× zoom (0.033 µm/pixel), 2048 × 2048 pixels, and in z-stacks with 0.37 µm steps. Due to the low penetration of the vGAT antibody, and following similar studies (*Heise et al., 2016*), these stacks were thinner (1.1–1.2 µm total depth) to capture most of the signal and perform reliable co-localisation assessment.

### Analysis

Puncta detection and dendritic tracing in three-dimensional (3D) stacks were performed using Imaris (version 9.1.2). Individual stacks were carefully examined in 3D by rotating the images and adjusting the zoom to ensure accurate identification of neuronal processes belonging to specific neurons. Cells for which this could not be confirmed were excluded from the analysis. Synaptophysin-mRuby belonging to specific GFP processes was detected based on their signal overlap with GFP. To automate puncta detection, synaptophysin-mRuby and gephyrin puncta were detected by estimating a punctum size of 0.5 µm and then using the 'Quality' filter. Detection threshold was kept within an unbiased consistent range (1000–2700) that allowed adaptation to variability between staining experiments and mice. Dendritic traces were performed using the 'Filaments' function, and the total length and maximum length of dendrites were calculated accordingly. Puncta density was estimated by dividing the total number of puncta by the total length of dendrites.

For co-localisation analysis, the 3D centroid coordinates for the synaptophysin-mRuby puncta detected in Imaris were exported and analysed using a custom-written script in MATLAB. This determined, at the y- and z-axis centroid co-ordinates of each mRuby punctum, the vGAT fluorescence intensity averaged over a 1 µm window centred on the local maximum of the mRuby distribution's x-position. If this mRuby-centred vGAT fluorescence intensity was greater than 1 standard deviation higher than local background vGAT fluorescence (taken from a 1 µm window immediately outside the x-axis border of the mRuby punctum), the punctum was considered to be 'co-localised'. The overall percentage of co-localised puncta was then determined for each cell.

For soma size measurements, 3D stacks were first transformed into maximum intensity projections. We used the polygon drawing tool on Fiji (*Schindelin et al., 2012*) to draw the soma and obtained the area measurements using the 'measure' function.

### Illustration

All images in the manuscript displaying examples of immunostaining were taken as high-resolution snapshots on ImarisViewer (version 10.2.0). Image stitching in *Figure 3* was performed using maximum intensity projections of individual stacks that were assembled using the ImageJ plugin MosaicJ (*Thévenaz and Unser, 2007*). Orthogonal views in *Figure 1F* were created in Fiji (*Schindelin et al., 2012*). Schematics, drawings, and figures layouts were designed using Inkscape (version 1.3.2).

## Acute slice preparation

DAT-tdT mice at P21–35 were decapitated under isoflurane anaesthesia, and the OB was removed and transferred into ice-cold slicing medium containing (in mM): 240 sucrose, 5 KCl, 1.25 $Na_2HPO_4$, 2

MgSO$_4$, 1 CaCl$_2$, 26 NaHCO$_3$, and 10 D-Glucose (all reagents from Sigma-Aldrich), bubbled with 95% O$_2$ and 5% CO$_2$. Horizontal slices (300 µm) of the OB were cut using a vibratome (VT1000S, Leica and 7000 smz-2, Campden Instruments) and maintained in artificial cerebrospinal fluid (ACSF) containing (in mM): 124 NaCl, 2.5 KCl, 1.25 Na$_2$HPO$_4$, 2 MgSO$_4$, 2 CaCl$_2$, 26 NaHCO$_3$, and 15 D-Glucose, bubbled with 95% O$_2$ and 5% CO$_2$. Slices were kept for 30 min in a water bath at 34°C and then left to recover for 30 min at RT.

## Acute slice electrophysiology

Individual slices were placed in a chamber mounted on an FN1 Fixed Stage Nikon microscope and held in place with a stainless-steel slice anchor. Slices were continuously superfused with ACSF heated to physiologically relevant temperature (34 ± 1°C) using an in-line solution heater (TC-344B, Warner Instruments). The ACSF contained, at all times, the NMDA receptor blocker DL-2-Amino-5-phosphonopentanoic acid at 50 µM (APV; A5282, Sigma-Aldrich), the AMPA receptor blocker 2,3-dioxo-6-nitro-7-sulfamoyl-benzo[f]quinoxaline at 10 µM (NBQX; N183, Sigma-Aldrich). The sodium channel blocking tetrodoxin citrate (TTX; T-550, Alomone and 1069, Tocris) was used at 1 µM for all experiments, excluding action potential recordings. Block of GABA-A receptors was achieved by the application of SR-95531 gabazine (1262, Tocris) for 5 min in the bath at 10 µM and confirmed by the absence of IPSCs. D1-like and D2-like receptors were blocked by applying SKF 83566 hydrobromide for 5 min in the bath at 10 µM (1586, Tocris) and sulpiride at 10 µM (S8010, Sigma-Aldrich), respectively. Calcium channels were blocked by applying cadmium chloride (202908, Sigma) at 200 µM for 5 min in the bath. The gap junction blocker carbenoxolone disodium at 100 µM (3096, Tocris) and the NKCC1 blocker bumetanide at 20 µM (3108, Tocris) were applied in the bath from the beginning and kept in the ACSF for at least 20–40 min. We adopted this strategy due to two key considerations: (1) carbenoxolone-induced instability in the membrane properties of the neurons during the initial minutes of application, and (2) sufficient time was required for chloride transporters to reach a steady intracellular chloride concentration ([Cl$^-$]) in response to bumetanide.

Recording pipettes were pulled from borosilicate glass (1B150F-4, World Precision Instruments) with a vertical puller (PC-10, Narishige). Some were tip fire-polished with a microforge (MF-830, Narishige). All AEI recordings were performed using caesium chloride intracellular solution (in mM: 125 CsCl, 10 HEPES, 0.1 EGTA, 4 MgATP, 0.3 Na$_3$GTP, 10 phosphocreatine, 10 GABA [280–290 mOsm and pHed to ~7.3 with CsOH], and pipette resistances of 2–4 MΩ).

### AEI recordings

DAT-tdT neurons were visualised with a 40× water immersion objective, and tdT fluorescence was revealed by LED excitation (CoolLED pE-100) with appropriate excitation and emission filters. All recorded cells were located in the OB's glomerular layer, including the border with the external plexiform layer. To ensure specificity in targeting only DA neurons, we avoided small, dimly fluorescent neurons which are likely to be of the calretinin-expressing glomerular subtype and excluded from our data any recordings showing evidence of those calretinin-positive cells' characteristically unstable and weak spiking (*Benito et al., 2018*; *Byrne et al., 2022*). Recordings were performed using a HEKA EPC 10/2 amplifier coupled to PatchMaster acquisition software (HEKA Elektronik). Signals were digitised and sampled at 20 kHz (50 µs interval sample) unless otherwise stated, and Bessel filtered at 10 kHz (filter 1) and 2.9 kHz (filter 2; active filters used in voltage clamp only).

Axon-bearing and anaxonic DA neurons were identified based on their action potential waveform. For that, cells were held at –60 mV in current clamp with 100% bridge balance. 10 ms somatic current injections of Δ+5 pA steps were applied until threshold was reached and an action potential occurred. 10 ms duration current injections were sampled at high temporal resolution 200 kHz (5 µs sample interval), and the recordings were smoothed prior to differentiation using a 20-point (100 µs) sliding filter. The first derivative of the smoothed trace was used for dV/dt and phase plane plot analyses. Neurons were then visually determined as axon-bearing or anaxonic based on their biphasic or monophasic phase plane plot profiles, respectively (*Galliano et al., 2018*).

Spontaneous IPSCs were recorded as inward events at –80 mV for 1 min while 1 µM TTX was added to the ACSF. Sodium channel block was confirmed by running an IV protocol and observing the block of the large, inward sodium current.

AEI recordings were sampled at 50 kHz (20 μs sample interval), with slow capacitance compensated. After membrane rupture, a test series was performed in voltage-clamp mode. The average current response to a double test pulse (5 ms steps, –10 and +10 mV) from the holding voltage of –60 mV was used to estimate series resistance ($R_S$; peak current), input resistance ($R_i$; steady-state holding current at new voltage), and membrane capacitance ($C_M$; area under the exponentially decaying current from peak to steady-state current). Series resistance ($R_S$) was assessed continuously and recordings were excluded if they exceeded 30 MΩ at any point or changed by more than 30% during any recording.

AEI protocols were performed in voltage-clamp from a holding potential of –80 mV, unless otherwise stated. Neurons were depolarised to 0 mV for 10 or 100 ms, before returning to –80 mV. Paired recordings were two 10 ms duration voltage steps with 50 ms inter-stimulus interval. All protocols were repeated three times, with 30 s in between each sweep, and an average response calculated. Due to the presence of passive currents, sweeps were analysed from 10 ms after the end of the voltage step for 500 ms. Gabazine and cadmium-sensitive AEI currents were measured as the difference between the pre- and the post-treatment averaged traces. The final charge was calculated as the area under the curve of the subtracted response. For AEI decay kinetics analysis, the time constant k was measured from a single exponential function fitted to the average response over the same time-frame used for charge calculation.

We confirmed no change in the AEI response over time by repeating the 10 ms voltage step protocol three times, with 5 min in between each protocol. No significant change in the charge of the AEI response was found during the experiment (*Figure 4—figure supplement 2A*; mean ± SEM: time-point 1, 25 ± 7.99 pC; timepoint 2, 27.29 ± 9.57 pC; timepoint 3, 29.29 ± 11.22 pC; n=5, N=4 mice; one-way repeated-measures ANOVA, p=0.44). We also ruled out the effect of somatic depolarisation on the AEI response by holding the DA neurons either at –50 mV or –80 mV for approximately 30 s before the start of the recordings. Neurons were then depolarised to 0 mV for 10 ms and then brought to –80 mV. The AEI response was not different between holding at –50 mV or –80 mV (*Figure 4—figure supplement 2B*; mean ± SEM: –50 mV response, 25.45 ± 4.53 pC; –80 mV response, 25.3 ± 7.3 pC; n=5, N=3 mice, paired t-test, p=0.96). These results suggest that our experimental design produces a reliable and stable response to study AEI in this neuronal population.

## Data analysis

When possible, experimenters were blinded to the conditions to perform the data analysis. For RNA sequencing data plotting, we made use of a whole OB single-cell RNA sequencing dataset (*Brann et al., 2020*), that integrates data from juvenile and adult mouse samples, contains labels for identified clusters for the different cell types, and for visualising the expression of genes of interest in identified cell types from the OB. Details about data processing, cluster inference, and annotation can be found in the original publication. Briefly, graph-based clustering was performed using the Louvain algorithm and cluster stability evaluated using Clustree. The clusters were annotated using known gene expression markers. Here, cluster 11 corresponds to the DA neurons and contained 1453 cells. These were identified by their expression of known OB DA neuron markers, including *Th*, *Trh*, *Pax6*, amongst others. Violin plots depicting the expression of *Th*, *Syp* (Synaptophysin), *Slc32a1* (vGAT), and the dopamine receptors *Ddr1-5*, in the cell cluster corresponding to the DA neurons, were generated with the R package Seurat. The dataset is available for download in the accompanying supplementary data, alongside the code used for generating the figures (see single-cell RNA sequencing of OB DA neurons: *Lipovsek and Grubb, 2020*, https://www.ncbi.nlm.nih.gov/geo/query/acc.cgi, GEO [Gene Expression Omnibus]; GSE151709).

Data analysis, statistical analysis, and plotting were carried out using GraphPad (San Diego, CA, USA), R, and custom-written MATLAB (Mathworks) scripts. Data are reported as mean ± SEM unless otherwise stated. 'N' represents the number of mice and 'n' is the number of cells. Details of individual statistical tests can be found in the Results section and in figure legends. Datasets were tested for normality using the D'Agostino and Pearson tests where the datasets were sufficiently large, and Shapiro-Wilk test if not. Statistical tests were determined considering the distribution of the data (normal or non-normal) and sample variance (equal or unequal). All tests were two-tailed, with α=0.05. Our experiments addressed questions of cellular morphology and function, and datasets comprised few neurons per animal, so comparisons were performed with cell as the unit of analysis. However, for all significant results, where false positive inflation due to pseudoreplicate data is of highest concern,

we additionally performed nested analyses that took intra-animal data dependence into account (*Aarts et al., 2014*). In all cases, these nested tests also returned significant differences between anaxonic and axon-bearing cell types.

## Acknowledgements

This work was supported by a Wellcome Trust Early-Career Award to AD-R and MSG (227621/Z/23/Z); BBSRC Research Grants (BB/V000195/1 and BB/N014650/1) and an H2020 ERC Consolidator Grant (FUNCOPLAN; 725729) to MSG; and an MRC Medical Research Council 4-year PhD studentship to DJB. We thank Elisa Galliano for valuable comments on the manuscript, the Andreae lab and the Centre for Developmental Neurobiology for image analysis support, Ikhlas Kechround and Nadine Leighton for immunostaining optimisation, the Burrone lab for advice on presynaptic labelling, Candida Tufo and Lorcan Browne for guidance and training in patch-clamp electrophysiology, Wing Ka Lo for mouse work support and the current members of the Grubb lab for insightful discussions throughout the project.

## Additional information

### Funding

| Funder | Grant reference number | Author |
| --- | --- | --- |
| Wellcome Trust | 10.35802/227621 | Ana Dorrego-Rivas<br>Matthew S Grubb |
| Biotechnology and Biological Sciences Research Council | BB/V000195/1 | Matthew S Grubb |
| Biotechnology and Biological Sciences Research Council | BB/N014650/1 | Matthew S Grubb |
| European Research Council | 10.3030/725729 | Matthew S Grubb |
| Medical Research Council | PhD studentship | Darren J Byrne |

The funders had no role in study design, data collection and interpretation, or the decision to submit the work for publication. For the purpose of Open Access, the authors have applied a CC BY public copyright license to any Author Accepted Manuscript version arising from this submission.

### Author contributions

Ana Dorrego-Rivas, Data curation, Formal analysis, Supervision, Funding acquisition, Investigation, Methodology, Writing – original draft, Project administration, Writing – review and editing; Darren J Byrne, Conceptualization, Data curation, Formal analysis, Funding acquisition, Investigation, Methodology, Project administration, Writing – review and editing; Yunyi Liu, Formal analysis, Investigation, Methodology; Menghon Cheah, Investigation, Methodology, Writing – review and editing; Ceren Arslan, Formal analysis, Investigation, Methodology, Writing – review and editing; Marcela Lipovsek, Data curation, Formal analysis, Methodology, Writing – review and editing; Marc C Ford, Methodology, Writing – review and editing; Matthew S Grubb, Conceptualization, Resources, Formal analysis, Supervision, Funding acquisition, Investigation, Methodology, Writing – original draft, Project administration, Writing – review and editing

### Author ORCIDs

Ana Dorrego-Rivas  https://orcid.org/0000-0002-0462-2988
Marcela Lipovsek  https://orcid.org/0000-0001-9328-0328
Marc C Ford  https://orcid.org/0000-0003-0472-2652
Matthew S Grubb  https://orcid.org/0000-0002-2673-274X

## Ethics

All experiments were performed in accordance with the United Kingdom Animals (Scientific Procedures) Act 1986, under the auspices of Home Office personal and project licenses held by the authors.

Reviewer #1 (Public review): https://doi.org/10.7554/eLife.105271.3.sa1
Reviewer #2 (Public review): https://doi.org/10.7554/eLife.105271.3.sa2
Author response https://doi.org/10.7554/eLife.105271.3.sa3

# Additional files

## Supplementary files

MDAR checklist

## Data availability

The full raw dataset supporting this article is openly available at the King's Open Research Data Repository as of the date of publication of the Version of Record (https://doi.org/10.18742/27731157).

The following dataset was generated:

| Author(s) | Year | Dataset title | Dataset URL | Database and Identifier |
|---|---|---|---|---|
| Dorrego-Rivas A, Byrne DJ, Liu Y, Cheah M, Arslan C, Lipovsek M, Ford MC, Grubb MS | 2025 | Strikingly different neurotransmitter release strategies in dopaminergic subclasses | https://doi.org/10.18742/27731157 | KCL figshare, 10.18742/27731157 |

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
