## [Editor Report · eLife Assessment]

This study provides evidence for distinct neurotransmitter release modalities between two subclasses of dopaminergic neurons in the olfactory bulb. Specifically, it demonstrates dendritic neurotransmitter release in anaxonic neurons and axonal release in axon-bearing neurons. The presence of GABAergic self-inhibition in anaxonic neurons further underscores the functional divergence between these subtypes. Overall, the manuscript presents **solid** evidence and offers biologically **important** insights into the organization and function of dopaminergic circuits within the olfactory bulb.

---

## [Referee Report · Reviewer #1 (Public review)]

Summary:

Dorrego-Rivas et al. investigated two different DA neurons and their neurotransmitter release properties in the main olfactory bulb. They found that the two different DA neurons in mostly glomerular layers have different morphologies as well as electrophysiological properties. The anaxonic DA neurons are able to self-inhibit but the axon-bearing ones are not. The findings are interesting and important to increase the understanding both of the synaptic transmissions in the main olfactory bulb and the DA neuron diversity. However, there are some major questions that the authors need to address to support their conclusions.

(1) It is known that there are two types of DA neurons in the glomerular layer with different diameters and capacitances (Kosaka and Kosaka, 2008; Pignatelli et al., 2005; Angela Pignatelli and Ottorino Belluzzi, 2017). In this manuscript, the authors need to articulate better which layer the imaging and ephys recordings took place, all glomerular layers or with an exception. Meanwhile, they have to report the electrophysiological properties of their recordings, including capacitances, input resistance, etc.

(2) It is understandable that recording the DA neurons in the glomerular layer is not easy. However, the authors still need to increase their n's and repeat the experiments at least three times to make their conclusion more solid. For example (but not limited to), Fig 3B, n=2 cells from 1 mouse. Fig.4G, the recording only has 3 cells.

(3) The statistics also use pseudoreplicates. It might be better to present the biology replicates, too.

(4) In Figure 4D, the authors report the values in the manuscript. It is recommended to make a bar graph to be more intuitive.

(5) In Figure 4F and G, although the data with three cells suggest no phenotype, the kinetics looked different. So, the authors might need to explore that aside from increasing the n.

(6) Similarly, for Figure 4I and J, L and M, it is better to present and analyze it like F and G, instead of showing only the after-antagonist effect.

Comments on revisions:

In the rebuttal, the authors argued that it had been extremely hard to obtain recordings stable enough for before-and-after effects on the same cell. Alternatively, they could perform the before-and-after comparison on different cells.

---

## [Referee Report · Reviewer #2 (Public review)]

Summary:

This study provides novel insights into the neurotransmitter release mechanisms employed by two distinct subclasses of dopaminergic neurons in the olfactory bulb (OB). The findings suggest that anaxonic neurons primarily release neurotransmitters through their dendrites, whereas axon-bearing neurons predominantly release neurotransmitters via their axons. Furthermore, the study reveals that anaxonic neurons exhibit self-inhibitory behavior, indicating that closely related neuronal subclasses may possess specialized roles in sensory processing.

Strengths:

This study introduces a novel and significant concept, demonstrating that two closely related neuron subclasses can exhibit distinct patterns of neurotransmitter release. Therefore, this finding establishes a valuable framework for future investigations into the functional diversity of neuronal subclasses and their contributions to sensory processing. Furthermore, these findings offer fundamental insights into the neural circuitry of the olfactory bulb, enhancing our understanding of sensory information processing within this critical brain region.

Weaknesses:

The reliance on synaptophysin-based presynaptic structures raises minor concerns about whether these structures represent functional synapses.

Comments on revisions:

Most of the concerns have been addressed by the authors, and there are no further comments about this manuscript.

---

## [Author Response]

The following is the authors’ response to the original reviews.

**Reviewer #1 (Public review):**
This Reviewer was positive about the study, stating ‘The findings are interesting and important to increase the understanding both of the synaptic transmissions in the main olfactory bulb and the DA neuron diversity.’ They provided a number of helpful suggestions for improving the paper, which we have incorporated as follows:(1) It is known that there are two types of DA neurons in the glomerular layer with different diameters and capacitances (Kosaka and Kosaka, 2008; Pignatelli et al., 2005; Angela Pignatelli and Ottorino Belluzzi, 2017). In this manuscript, the authors need to articulate better which layer the imaging and ephys recordings took place, all glomerular layers or with an exception. Meanwhile, they have to report the electrophysiological properties of their recordings, including capacitances, input resistance, etc.

We thank the Reviewer for this clarification. Indeed, the two dopaminergic cell types we study here correspond directly to the subtypes previously identified based on cell size. Our previous work showed that axon-bearing OB DA neurons have significantly larger somas than their anaxonic neighbours (Galliano et al. 2018), and we replicate this important result in the present study (Figure 3D). In terms of electrophysiological correlates of cell size, we now provide full details of passive membrane properties in the new Supplementary Figure 4, as requested. Axon-bearing DA neurons have significantly lower input resistance and show a non-significant trend towards higher cell capacitance. Both features are entirely consistent with the larger soma size in this subtype. We apologise for the oversight in not fully describing previous categorisations of OB DA neurons, and have now added this information and the appropriate citations to the Introduction (lines 56 to 59 of the revised manuscript).

In terms of cell location, all cells in this study were located in the OB glomerular layer. We sampled the entire glomerular layer in all experiments, including the glomerular/EPL border where the majority of axon-bearing neurons are located (Galliano et al. 2018). This is now clarified in the Materials and Methods section (lines 535 to 537 and 614 to 616 of the revised manuscript).

(2) It is understandable that recording the DA neurons in the glomerular layer is not easy. However, the authors still need to increase their n's and repeat the experiments at least three times to make their conclusion more solid. For example (but not limited to), Fig 3B, n=2 cells from 1 mouse. Fig.4G, the recording only has 3 cells.

Despite the acknowledged difficulty of these experiments, we have now added substantial extra data to the study as requested. We have increased the number of cells and animals to further support the following findings:

Fig 3B: we now have n=5 cells from N=3 mice. We have created a new Supplementary Figure 1 to show all the examples.

Figure 4G: we now have n=6 cells from N=4 mice.

Figure 5G: we now have n=3 cells from N=3 mice.

The new data now provide stronger support for our original conclusions. In the case of auto-evoked inhibition after the application of D1 and D2 receptor antagonists, a nonsignificant trend in the data suggests that, while dopamine is clearly not necessary for the response, it may play a small part in its strength. We have now included this consideration in the Results section (lines 256 to 264 of the revised manuscript).

(3) The statistics also use pseudoreplicates. It might be better to present the biology replicates, too.

Indeed, in a study focused on the structural and functional properties of individual neurons, we performed all comparisons with cell as the unit of analysis. This did often (though not always) involve obtaining multiple data points from individual mice, but in these low-throughput experiments n was never hugely bigger than N. The potential impact of pseudoreplicates and their associated within-animal correlations was therefore low. We checked this in response to the Reviewer’s comment by running parallel nested analyses for all comparisons that returned significant differences in the original submission. These are the cases in which we would be most concerned about potential false positive results arising from intra-animal correlations, which nested tests specifically take into account (Aarts et al., 2013). In every instance we found that the nested tests also reported significant differences between anaxonic and axonbearing cell types, thus fully validating our original statistical approach. We now report this in the relevant section of the Materials and Methods (lines 686 to 691 of the revised manuscript).

(4) In Figure 4D, the authors report the values in the manuscript. It is recommended to make a bar graph to be more intuitive.

This plot does already exist in the original manuscript. We originally describe these data to support the observation that an auto-evoked inhibition effect exists in anaxonic neurons (corresponding to now lines 240 to 245 of the revised manuscript). We then show them visually in their entirety when we compare them to the lack of response in axon-bearing neurons, depicted in Figure 5C. We still believe that this order of presentation is most appropriate for the flow of information in the paper, so have maintained it in our revised submission.

(5) In Figure 4F and G, although the data with three cells suggest no phenotype, the kinetics looked different. So, the authors might need to explore that aside from increasing the n.

We thank the Reviewer for this suggestion. To quantify potential changes in the autoevoked inhibition response kinetics, we fitted single exponential functions and compared changes in the rate constant (k; Methods, lines 650 to 652 of the revised manuscript). Overall, we observed no consistent or significant change in rate constant values after adding DA receptor antagonists. This finding is now reported in the Results section (lines 260 to 263 of the revised manuscript) and shown in a new Supplementary Figure 3.

(6) Similarly, for Figure 4I and J, L and M, it is better to present and analyze it like F and G, instead of showing only the after-antagonist effect.

We agree that the ideal scenario would have been to perform the experiments in Figure 4J and 4M the same way as those in Figure 4G, with a before vs after comparison. Unfortunately, however, this was not practically possible.

When attempting to apply carbenoxelone to already-patched cells, we found that this drug highly disrupted the overall health and stability of our recordings immediately after its application. This is consistent with previous reports of similar issues with this compound (e.g. Connors 2012, Epilepsy Currents; Tovar et al., 2009, Journal of Neurophysiology). After many such attempts, the total yield of this experiment was one single cell from one animal. Even so, as shown in the traces below, we were able to show that the auto-evoked inhibition response was not eliminated in this specific case:

**Author response image 1. sa3fig1:** Traces of an AEI response recorded before (magenta) and after (green) the application of carbenoxolone (n=1 cell from N=1 mouse).

In light of these issues, we instead followed published protocols in applying the carbenoxolone directly in the bath without prior recording for 20 minutes (following Samailova et al., 2003, Journal of Neurochemistry) and ran the protocol after that time. Given that our main question was to ask whether gap junctions were strictly necessary for the presence of any auto-evoked inhibition response, our positive findings in these experiments still allowed us to draw clear conclusions.

In contrast, the issue with the NKCC1 antagonist bumetanide was time. As acknowledged by this Reviewer, obtaining and maintaining high-quality patch recordings from OB DA neurons is technically challenging. Bumetanide is a slow-acting drug when used to modify neuronal chloride concentrations, because in addition to the time it takes to reach the neurons and effectively block NKCC1, the intracellular levels of chloride subsequently change slowly. Studies using this drug in slice physiology experiments typically use an incubation time of at least 20 minutes (e.g. Huberfeld et al., 2007, Journal of Neuroscience), which was incompatible with productive data collection in OB DA neurons. Again, after many unsuccessful efforts, we were forced instead to include bumetanide in the bath without prior recording for 20-30 minutes. As with the carbenoxolone experiment, our goal here was to establish whether autoevoked inhibition was in any way retained in the presence of this drug, so our positive result again allowed us to draw clear conclusions.

**Reviewer #1 (Recommendations for the authors):**
(1) I suggest the authors reconsider the terminology. For example, they use "strikingly" in their title. The manuscript reported two different transmitter release strategies but not the mechanisms, and the word "strikingly" is not professional, either.

We appreciate the Reviewer’s attention to clarity and tone in the manuscript title, and have nevertheless decided to retain the original wording. The almost all-or-nothing differences between closely related cell types shown in structural and functional properties here (Figures 3F & 5C) are pronounced, extremely clear and easily spotted – all properties appropriate for the word ‘striking.’ In addition, we note that the use of this term is not at all unprofessional, with a PubMed search for ‘strikingly’ in the title of publications returning over 200 hits.

(2) Similarly, almost all confocal scopes are 3D because images can be taken at stacks. So "3D confocal" is misleading.

We understand that this is misleading. We have now replaced the sentence ‘Example snapshot of a 3D confocal stack of…’ by ‘Example confocal images of…’ in all the figure legends that apply.

(3) It is recommended to present the data in bar graphs with data dots instead of showing the numbers in the manuscript directly.

We agree entirely, and now present data plots for all comparisons reported in the study (Supplementary Figures 2, 4 and 5).

**Reviewer #2 (Recommendations for the authors):**
(1) Several experiments report notably small sample sizes, such as in Figures 3B and 5G, where data from only 2 cells derived from 1-2 mice are presented. Figures 4E-G also report the experimental result only from 3 cells derived from 3 mice. To enhance the statistical robustness and reliability of the findings, these experiments should be replicated with larger sample sizes.

As per our response to Reviewer 1’s comment #2 above, and to directly address the concern that some evidence was ‘incomplete’, we have now added significant extra data and analysis to this revised submission (Figures 4 and 5; and Supplementary Figure 1). We believe that this has further enhanced the robustness and reliability of our findings, as requested.

(2) The authors utilize vGAT-Cre for Figures 1-3 and DAT-tdTomato for Figures 4-5, raising concerns about consistency in targeting the same population of dopaminergic neurons. It remains unclear whether all OB DA neurons express vGAT and release GABA. Clarification and additional evidence are needed to confirm whether the same neuronal population was studied across these experiments.

Although we indeed used different mouse lines to investigate structural and functional aspects of transmitter release, we can be very confident that both approaches allowed us to study the same two distinct DA cell types being compared in this paper. Existing data to support this position are already clear and strong, so in this revision we have focused on the Reviewer’s suggestion to clarify the approaches we chose.

First, it is well characterised that in mouse and many other species all OB DA neurons are also GABAergic. This has been demonstrated comprehensively at the level of neurochemical identity and in terms of dopamine/GABA co-release, and is true across both small-soma/anaxonic and large-soma/axon-bearing subclasses (Kosaka & Kosaka 2008; 2016; Maher & Westbrook 2008; Borisovska et al., 2013; Vaaga et al., 2016; Liu et al. 2013). To specifically confirm vGAT expression, we have also now provided additional single-cell RNAseq data and immunohistochemical label in a revised Figure 1 (see also Panzanelli et al., 2007, now referenced in the paper, who confirmed endogenous vGAT colocalisation in TH-positive OB neurons). Most importantly, by using vGAT-cre mice here we were able to obtain sufficient numbers of both anaxonic and axon-bearing DA neurons among the vGAT-cre-expressing OB population. We could unambiguously identify these cells as dopaminergic because of their expression of TH protein which, due to the absence of noradrenergic neurons in the OB, is a specific and comprehensive marker for dopaminergic cells in this brain region (Hokfelt et al., 1975; Rosser et al., 1986; Kosaka & Kosaka 2016). Crucially, both axon-bearing and anaxonic OB DA subtypes strongly express TH (Galliano et al., 2018, 2021). We have now added additional text to the relevant Results section (lines 99 to 108 of the revised manuscript) to clarify these reasons for studying vGAT-cre mice here.

We were also able to clearly identify and sample both subtypes of OB DA neuron using DAT-tdT mice. Our previous published work has thoroughly characterised this exact mouse line at the exact ages studied in the present paper (Galliano et al., 2018; Byrne et al., 2022). We know that DAT-tdT mice provide rather specific label for TH-expressing OB DA neurons (75% co-localisation; Byrne et al., 2022), but most importantly we know which non-DA neurons are labelled in this mouse line and how to avoid them. All nonTH-expressing but tdT-positive cells in juvenile DAT-tdT mice are small, dimly fluorescent and weakly spiking neurons of the calretinin-expressing glomerular subtype (Byrne et al., 2022). These cells are easily detected during physiological recordings, and were excluded from our study here. This information is now provided in the relevant Methods section (lines 616 to 619 of the revised manuscript, also referenced in lines 236 to 240 of the results section), and we apologise for its previous omission. Finally, we have shown both structurally and functionally that both axon-bearing and anaxonic OB DA subtypes are labelled in DAT-tdT mice (Galliano et al., 2018, Tufo et al., 2025; present study). Overall, these additional clarifications firmly establish that the same neuronal populations were indeed studied across our experiments.

(3) The low TH+ signal in Figure 1D raises questions regarding the successful targeting of OB DA neurons. Further validation, such as additional staining, is required to ensure that the targeted neurons are accurately identified.

As noted in our response to the previous comment, TH is a specific marker for dopaminergic neurons in the mouse OB, and is widely used for this purpose. Labelling for TH in our tissue is extremely reliable, and in fact gives such strong signal that we were forced to reduce the primary antibody concentration to 1:50,000 to prevent bleedthrough into other acquisition channels. Even at this concentration it was extremely straightforward to unambiguously identify TH-positive cells based on somatic immunofluorescence. We recognise, however, that the original example image in Figure 1D was not sufficiently clear, and have now provided a new example which illustrates the TH-based identification of these cells much more effectively.

(4) Estimating the total number of dopaminergic neurons in the olfactory bulb, along with the relative proportions of anaxonic and axon-bearing neuron subtypes, would provide valuable context for the study. Presenting such data is crucial to underscore the biological significance of the findings.

This information has already been well characterised in previous studies. Total dopaminergic cell number in the OB is ~90,000 (Maclean & Shipley, 1988; Panzanelli et al., 2007; Parrish-Aungst et al., 2007). In terms of proportions, anaxonic neurons make up the vast majority of these cells, with axon-bearing neurons representing only ~2.5% of all OB dopaminergic neurons at P28 (Galliano et al., 2018). Of course, the relatively low number of the axon-bearing subtype does not preclude its having a potentially large influence on glomerular networks and sensory processing, as demonstrated by multiple studies showing the functional effects of inter-glomerular inhibition (Kosaka & Kosaka, 2008; Liu et al., 2013; Whitesell et al., 2013; Banerjee et al., 2015). This information has now been added to the Introduction (line 47 and lines 59 to 62 of the revised manuscript).

(5) The authors report that in-utero injection was performed based on the premise that the two subclasses of dopaminergic neurons in the olfactory bulb are generated during embryonic development. However, it remains unclear whether in-utero injection is essential for distinguishing between these two subclasses. While the manuscript references a relevant study, the explanation provided is insufficient. A more detailed justification for employing in-utero injection would enhance the manuscript's clarity and methodological rigor.

We apologise for the lack of clarity in explaining the approach. In utero injection is not absolutely essential for distinguishing between the two subclasses, but it does have two major advantages. (1) Because infection happens before cells migrate to their final positions, it produces sparse labelling which permits later unambiguous identification of individual cells’ processes; and (2) Because both subclasses are generated embryonically (compared to the postnatal production of only anaxonic DA neurons), it allows effective targeting of both cell types. We have now expanded the relevant section of the Results to explain the rationale for our approach in more detail (lines 109 to 116 of the revised manuscript).

(6) In Figures 1A and 4A, it appears that data from previously published studies were utilized to illustrate the differential mRNA expression in dopaminergic neurons of the olfactory bulb. However, the Methods section and the manuscript lack a detailed description of how these dopaminergic neurons were classified or analyzed. Given that these figures contribute to the primary dataset, providing additional explanation and context is essential to ensure clarity of the findings.

We apologise for the lack of clarity. We have now extended the part of the methods referring to the RNAseq data analysis (lines 666 to 678 of the revised manuscript).

(7) In Figure 2C, anaxonic dopamine neurons display considerable variability in the number of neurotransmitter release sites, with some neurons exhibiting sparse sites while others exhibit numerous sites. The authors should address the potential biological or methodological reasons for this variability and discuss its significance.

We thank the Reviewer for highlighting this feature of our data. We have now outlined potential methodological reasons for the variability, whilst also acknowledging that it is consistent with previous reports of presynaptic site distributions in these cells (Kiyokage et al., 2017; Results, lines 169 to 172 of the revised manuscript). We have also added a brief discussion of the potential biological significance (Discussion, lines 446 to 450).

(8) In the images used to differentiate anaxonic and axon-bearing neurons, the soma, axons, and dendrites are intermixed, making it difficult to distinguish structures specific to each subclass. Employing subclass-specific labeling or sparse labeling techniques could enhance clarity and accuracy in identifying these structures.

Distinguishing these structures is indeed difficult, and was the main reason we used viral label to produce sparse labelling (see response to comment #5 above). In all cases we were extremely careful, including cells only when we could be absolutely certain of their anaxonic or axon-bearing identity, and could also be certain of the continuity of all processes. Crucially, while the 2D representations we show in our figures may suggest a degree of intermixing, we performed all analyses on 3D image stacks, significantly improving our ability to accurately assign structures to individual cells. We have now added extra descriptions of this approach in the relevant Methods section (lines 546 to 548 of the revised manuscript).

(9) In Figure 3, the soma area and synaptophysin puncta density are compared between axon-bearing and anaxonic neurons. However, the figure only presents representative images of axon-bearing neurons. To ensure a fair and accurate comparison, representative images of both neuron subtypes should be included.

The original figures did include example images of puncta density (or lack of puncta) in both cell types (Figure 2B and Figure 3E). For soma area, we have now included representative images of axon-bearing and anaxonic neurons with an indication of soma area measurement in a new Supplementary Figure 2A.

(10) In Figure 4B, the authors state that gephyrin and synaptophysin puncta are in 'very close proximity.' However, it is unclear whether this proximity is sufficient to suggest the possibility of self-inhibition. Quantifying the distance between gephyrin and synaptophysin puncta would provide critical evidence to support this claim. Additionally, analyzing the distribution and proportion of gephyrinsynaptophysin pairs in close proximity would offer further clarity and strengthen the interpretation of these findings.

We thank the Reviewer for raising this issue. We entirely agree that the example image previously shown did not constitute sufficient evidence to claim either close proximity of gephyrin and synaptophysin puncta, nor the possibility of self-inhibition. We are not in a position to perform a full quantitative analysis of these spatial distributions, nor do we think this is necessary given previous direct evidence for auto-evoked inhibition in OB dopaminergic cells (Smith and Jahr, 2002; Murphy et al., 2005; Maher and Westbrook, 2008; Borisovska et al., 2013) and our own demonstration of this phenomenon in anaxonic neurons (Figure 4). We have therefore removed the image and the reference to it in the text.

(11) In Figures 4J and 4M, the effects of the drugs are presented without a direct comparison to the control group (baseline control?). Including these baseline control data is essential to provide a clear context for interpreting the drug effects and to validate the conclusions drawn from these experiments.

We appreciate the Reviewer’s attention to this important point. As this concern was also raised by Reviewer 1 (their point #6), we have provided a detailed response fully addressing it in our replies to Reviewer 1 above.

(12) In Lines 342-344, the authors claim that VMAT2 staining is notoriously difficult. However, several studies (e.g., Weihe et al., 2006; Cliburn et al., 2017) have successfully utilized VMAT2 staining. Moreover, Zhang et al., 2015 - a reference cited by the authors - demonstrates that a specific VMAT2 antibody effectively detects VMAT2. Providing evidence of VMAT2 expression in OB DA neurons would substantiate the claim that these neurons are GABA-co-releasing DA neurons and strengthen the study's conclusions.

As noted in response to this Reviewer’s comment #2 above, there is clear published evidence that OB DA neurons are GABA- and dopamine-releasing cells. These cells are also known to express VMAT2 (Cave et al., 2010; Borisovska et al., 2013; Vergaña-Vera et al., 2015). We do not therefore believe that additional evidence of VMAT2 expression is necessary to strengthen our study’s conclusions. We did make every effort to label VMAT2-positive release sites in our neurons, but unfortunately all commercially available antibodies were ineffective. The successful staining highlighted by the Reviewer was either performed in the context of virally driven overexpression (Zhang et al., 2015) or was obtained using custom-produced antibodies (Weihe et al., 2006; Cliburn et al., 2017). We have now modified the Discussion text to provide more clarification of these points (lines 393 to 395 of the revised manuscript).